# Provably Efficient Linear Bandits with Instantaneous Constraints in Non-Convex Feature Spaces

## Abstract

In linear stochastic bandits, tasks with instantaneous hard constraints present significant challenges, particularly when the feature space is non-convex or discrete. This is especially relevant in applications such as financial management, recommendation systems, and medical treatment selection, where safety constraints appear in non-convex forms or where decisions must often be made within non-convex and discrete sets. In these systems, bandit methods rely on the ability of feature functions to extract critical features. However, in contrast to the star-convexity assumption commonly discussed in the literature, these feature functions often lead to non-convex and more complex feature spaces. In this paper, we investigate linear bandits and introduce a method that operates effectively in a non-convex feature space while satisfying instantaneous hard constraints at each time step. We demonstrate that our method, with high probability, achieves a regret of $\tilde{\mathcal{O}}\big(d(1+\frac{\tau}{\epsilon\iota})\sqrt{T}\big)$ and meets the instantaneous hard constraints, where $d$ represents the feature space dimension, $T$ the total number of rounds, and $\tau$ a safety related parameter. The constant parameters $\epsilon$ and $\iota$ are related to our localized assumptions around the origin and the optimal point. In contrast, standard safe linear bandit algorithms that rely on the star-convexity assumption often result in linear regret. Furthermore, our approach handles discrete action spaces while maintaining a comparable regret bound. Moreover, we establish an information-theoretic lower bound on the regret of $\Omega\left(\max\{d\sqrt{T},\frac{1}{\epsilon\iota^2}\}\right)$ for $T \geq \frac{32e}{\epsilon\iota^2}$, emphasizing the critical role of $\epsilon$ and $\iota$ in the regret upper bound. Lastly, we provide numerical results to validate our theoretical findings.

## 1 Introduction

The linear bandit (LB) problem is a framework in decision theory and machine learning designed to address real-world scenarios with large, and potentially uncountable, decision sets (Abbasi-Yadkori et al., 2011; Russo & Van Roy, 2014; Soare et al., 2014). In this setting, the expected reward for an action (or "arm") is modeled as the inner product between a feature vector and an unknown parameter. To maximize cumulative reward over a sequence of trials, an agent must balance two competing objectives: exploration, where actions are chosen to estimate this unknown parameter, and exploitation, where the agent uses the estimation to select actions that yield high rewards. Striking the right balance between exploration and exploitation is key to optimizing rewards over time.

Many real-world applications impose strict limitations that require instantaneous hard constraints to be satisfied at every time step Shi et al. (2023). For instance, in resource allocation, resource constraints must be met in real-time to avoid stockouts or logistical failures. Similarly, in AI-driven medical treatments, decisions must consistently prioritize safety (Xiong et al., 2024; Vamvoudakis et al., 2021; Thomas et al., 2019). This work aims to address the problem of LB under instantaneous hard constraints, specifically in **non-convex and discrete feature spaces**. Earlier studies, such as Amani et al. (2019); Moradipari et al. (2021); Pacchiano et al. (2024), have shown that near-optimal performance can be achieved in linear bandits with instantaneous hard constraints in *convex or star-convex* feature spaces respectively. In these approaches, the agent initially constructs a conservative

estimated safe set and begins interacting with the environment by sampling from this set. It then gradually expands the estimated safe set toward the true safe set as it gathers more experience.

One might ask why we are interested in problems with non-convex and discrete feature spaces. The reason is that many real-world applications inherently involve structures that convex and star-convex assumptions fail to capture. Applications such as financial management, recommendation systems, and medical treatment selection often involve action sets that are neither convex nor star-convex, but rather composed of discrete or separated subsets (see Section 5.2). Additionally, non-convexity frequently arises in modern machine learning problems due to the use of function approximators with non-linear feature functions, such as Deep Neural Networks (DNNs), Radial Basis Functions (RBFs), and Fourier basis features (Sutton & Barto, 2018; Zhu et al., 2023; Mnih et al., 2016; Kalashnikov et al., 2018). In DNNs, for example, non-convex activation functions like ReLU, Sigmoid, and Tanh contribute to the overall non-convexity of the feature space.

A question arises: can the same conservative strategy used in convex and star-convex settings, as discussed in Amani et al. (2021); Pacchiano et al. (2024), be directly applied to our case? The answer is **no**, as applying this approach in non-convex spaces is not straightforward. In fact, conservative strategies in such settings may introduce a bias toward suboptimal directions, leading to linear regret, as shown in Fig. 1a. (See Section 6 for a complete description of the simulation.) We refer to this issue as **non-convexity bias**.

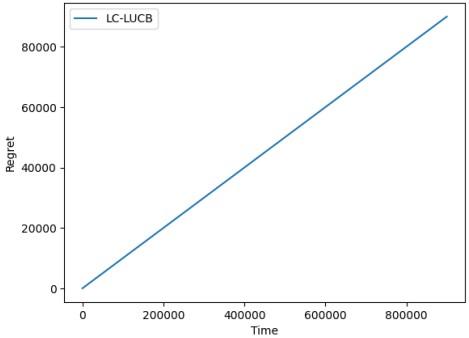
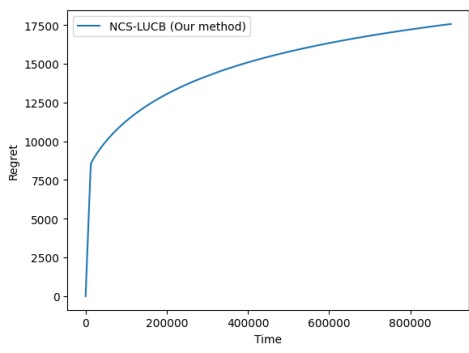

(a) Regret for LC-LUCB (Pacchiano et al., 2024).      (b) Regret for NCS-LUCB (ours).

Figure 1: Comparison of the average regret for NCS-LUCB (our method) and LC-LUCB in Pacchiano et al. (2024) over 10 trials.

To understand **non-convexity bias**, consider the following example: a LB problem with an action set $\mathcal{A} = \{a_1, a_2, a_3, a_4\}$ where $a_3$ is the optimal action as illustrated in Fig. 2a. Assume that the agent initially knows the actions in the rectangle $\mathcal{R}$ are safe, specifically $\{a_1, a_2\}$, are safe but cannot verify the safety of $\{a_3, a_4\}$.

The core idea of the conservative strategy in LB, as proposed by Amani et al. (2021); Pacchiano et al. (2024) is that the agent can gather noisy information about the cost of $a_3$ by playing action $a_1$, as both lie along the same direction, i.e., the $x$-axis. Similarly, the agent can estimate the cost of $a_4$ by playing action $a_2$, as both are aligned along the $y$-axis. Thus, a UCB-based bonus is used to ensure the agent explores both the $x$- and $y$-directions by playing $a_1$ and $a_2$ enough times, eventually expanding the safe set to include the optimal action $a_3$.

In Amani et al. (2021); Pacchiano et al. (2024), the bonus term for $a_1$ is designed based on the distance between $a_3$ and the current safe set's boundary along the $x$-axis. In the context of our problem, the bonus term in Amani et al. (2021); Pacchiano et al. (2024) is calculated based on the distance between $a_3$ and the rectangle $\mathcal{R}$, i.e., $d_1$, mistakenly assuming $a_1$ lies on $\mathcal{R}$. However, as depicted in Fig. 2a, $a_1$ is far from $\mathcal{R}$, and the distance between $a_1$ and $a_3$, i.e., $d_2$, is significantly larger than $d_1$. As a result, the bonus for $a_1$ is not large enough to incentivize the agent to play $a_1$, potentially biasing it toward playing $a_2$ instead. Consequently, the agent may not explore the $x$-direction sufficiently to estimate the cost of $a_3$ and verify its safety, leading to linear regret.

To resolve this issue, the correct bonus should be based on $d_2$. This problem does not arise in convex or star-convex settings, as these structures always ensure that $d_1 = d_2$ since all the points connecting $a_1$ and $a_3$ are in the action set. For a more detailed discussion on this bias, refer to Section 5.2.

**Our contribution.** In this work, we make the first attempt to *design near-optimal safe algorithms for linear bandit problems with instantaneous hard constraints in non-star-convex and discrete spaces*. In these problems, the reward and costs associated with each action $a$ are modeled as linear functions of a known, fixed feature mapping $\phi(a)$, where $\phi : \mathcal{A} \to \mathbb{R}^d$ (Amani et al., 2019; Pacchiano et al., 2024; Moradipari et al., 2021). We summarize our main contributions below:

**1.** We propose an algorithm, *Non-Convex Safe Linear UCB* (NCS-LUCB), for linear bandit problems with non-convex feature spaces under instantaneous hard constraints. NCS-LUCB achieves a regret of $\tilde{\mathcal{O}}\big(d(1 + \frac{1}{\tau\epsilon\iota})\sqrt{T}\big)$ with high probability, nearly matching the regret bounds in convex and star-convex settings while ensuring safety at each step. These non-convex spaces adhere to specific local assumptions around the initially known safe action and feature points near the constraint boundary, as outlined in Assumption 3. Here, $d$ is the feature space dimension, $T$ is the total number of rounds, and $\tau$ is a safety-related parameter. The bounded constants $\epsilon$ and $\iota$ are related to our local assumptions around the origin and the optimal point. *To the best of our knowledge, this is the first result for non-convex and discrete settings under such local assumptions. In Appendix J, we show that our result also obtain the same regret bound for the linear contextual bandit without assuming star convexity, thus, extending the result of Pacchiano et al. (2024).*

**2.** We provide a lower bound on the regret of $\Omega\left(\max\{d\sqrt{T}, \frac{1}{\epsilon\iota^2}\}\right)$ for this problem, highlighting the necessity of $\epsilon$ and $\iota$ in the upper bound. This also implies that Assumption 3 cannot be further relaxed.

**3.** To address the non-convexity bias, we introduce a new bonus term in Section 4, ensuring that the agent explores beyond suboptimal directions. This bonus is intentionally more optimistic than those designed for convex and star-convex cases to maintain the optimism property in non-convex spaces (see Lemma 2). Despite this increased optimism, the bonus still leads to sublinear regret (see Lemma 4).

**Related works.** Kazerouni et al. (2017) studied linear bandits under the constraint that the cumulative reward must exceed a baseline policy's performance with high probability. Amani et al. (2019) extended this to linear bandits with convex decision sets and stage-wise hard constraints, proposing a UCB-based method with two phases, achieving a regret of $\tilde{O}(T^{\frac{2}{3}})$. Moradipari et al. (2021) assumed star-convex decision sets, applying Thompson Sampling to achieve a regret of $\tilde{O}(d^{\frac{3}{2}}\frac{\sqrt{T}}{\tau})$. Pacchiano et al. (2021) examined the linear bandit problem under a slightly more relaxed condition, assuming the constraint is satisfied in expectation over the policy, rather than with high probability. This approach yielded a regret of $\tilde{O}(d\frac{\sqrt{T}}{\tau})$, and they also provided a lower bound to show that the dependence of the upper bound on $\frac{1}{\tau}$ is essential. Pacchiano et al. (2024) later showed similar results under high-probability constraints. Hutchinson et al. (2024) introduced "directional optimism" for linear bandits with instantaneous hard constraints, achieving improved regret for well-separated problem instances. Other related works include Gangrade et al. (2024); Afsharrad et al. (2024); Zhou & Ji (2022); Deng et al. (2022); Agrawal et al. (2016); Khezeli & Bitar (2020); Moradipari et al. (2020); Camilleri et al. (2022). For a more detailed discussion on related works, please refer to Appendix B.

## 2 PROBLEM FORMULATION

In this paper, we focus on a constrained bandit problem, denoted as $(\mathcal{A}, r, c)$, operating in an online setting over $T \in \mathbb{N}$ rounds. Here, $\mathcal{A}$ represents the action space, and $r$ and $c$ correspond to the reward function and cost function at each step, respectively.

During round $t \in [T]$, the learner interacts with the environment by selecting an action $a_t \in \mathcal{A}$. Subsequently, the learner observes a noisy reward $\hat{r}_t(a_t) = r(a_t) + \eta_t$, where $r(.) : \mathcal{A} \to [0,1]$ represents an unknown function, and $\eta_t$ denotes a zero-mean $\sigma$-sub-Gaussian random variable. In addition, it observes a corresponding noisy cost $\hat{c}_t(a_t) = c(a_t) + \zeta_t$, where $c(.) : \mathcal{A} \to [0,1]$ is an unknown cost function, and $\zeta_t$ is a zero-mean $\sigma$-sub-Gaussian random variable.

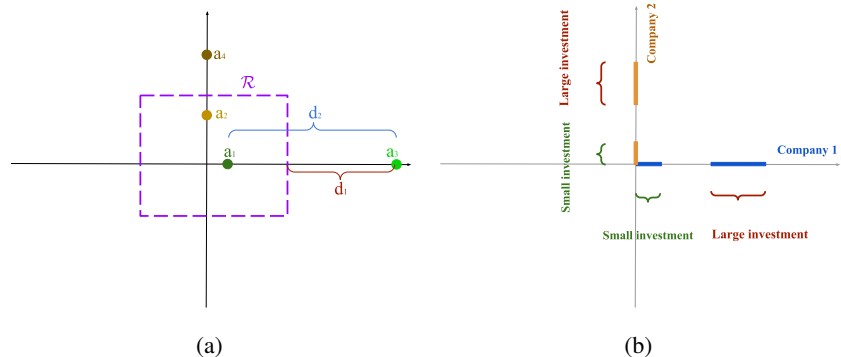

(a) (b)

Figure 2: **(a)** The bonus proposed in Amani et al. (2021); Pacchiano et al. (2024) is calculated based on the distance between the optimal point $a_3$ and the safety zone $\mathcal{R}$, but this bonus does not ensure sufficient exploration in the $x$-direction. A suitable bonus should be larger than $d_2$. **(b)** A VC can invest in different companies at various levels. This figure illustrates the case for two companies, but the concept can be extended to more.

**Notations.** For any vector $v \in \mathbb{R}^2$, the normalized vector is defined as $\overline{v} := \frac{v}{\|v\|}$, where $\|.\|$ denotes the $l_2$ norm. For any positive semi-definite matrix $A$, the operator $\|v\|_A$ defines the weighted norm as $\|v\|_A := \sqrt{v^T A v}$. For all $T \in \mathbb{N}$, $[T] \triangleq \{1, \ldots, T\}$. Also, for a mapping $f(.) : \mathbb{R}^m \to \mathbb{R}^n$ and a set $\mathcal{B} \subset \mathbb{R}^m$, we define $f(\mathcal{B}) \triangleq \{y \in \mathbb{R}^n \mid \exists b \in \mathcal{B} : y = f(b)\}$.

**Instantaneous hard constraint.** In each round $t$, the learner is required to adhere to a hard constraint: $c(a_t) \leq \tau$, where $\tau$ is a known positive constant that serves as the safety threshold. The corresponding safe action set is defined as $\mathcal{A}^{\text{safe}} \triangleq \{a \in \mathcal{A} : c(a) \leq \tau\}$.

**Performance metric.** Let $T$ represent the total number of rounds in which the agent interacts with the environment, and $\{a_t\}_{t=1}^T$ denote the actions selected by the agent during these rounds. The agent's performance is measured by regret as follows: $\text{Regret}(T) \triangleq \sum_{t=1}^T [r(a^*) - r(a_t)]$, where $a^*$ is the optimal action that maximizes the reward function $r(.)$ while satisfying the safety constraint, defined as $a^* \triangleq \arg\max_{a \in \mathcal{A}^{\text{safe}}} r(a)$.

**Linear bandits.** To handle the large and potentially infinite number of actions, we concentrate on linear bandits. This choice enables us to employ linear function approximation methods to solve our problem effectively.

**Assumption 1** *(Linear bandits (Amani et al., 2019; Pacchiano et al., 2024)) Consider a constrained bandit problem denoted as $(\mathcal{A}, r, c)$, which is assumed to be a linear bandit problem with a feature function $\phi : \mathcal{A} \to \mathcal{F} \subset \mathbb{R}^d$. Specifically, there exist unknown vectors $\theta^*$ and $\gamma^*$ in $\mathbb{R}^d$ such that for any $a \in \mathcal{A}$, the reward and cost functions are given by $r(a) = \langle \phi(a), \theta^* \rangle$ and $c(a) = \langle \phi(a), \gamma^* \rangle$, respectively. Additionally, we assume, without loss of generality, that for all $a \in \mathcal{A}$, we have $\|\phi(a)\| \leq L$ for some $L \in (0, 1]$, and $\max(\|\theta^*\|, \|\gamma^*\|) \leq \sqrt{d}$, where $d$ is the dimension of the feature space.*

Assumption 1 encapsulates the linear relationship between both the cost and reward functions and the feature map. It is important to note that, despite this linearity, the feature map $\phi(.)$ itself may be non-linear, and its image in the feature space can result in a non-convex space.

**Initial safe action.** Designing a safe bandit algorithm that achieves sublinear regret requires at least one known safe action, as shown in Theorem 3 of Shi et al. (2023). This assumption is often valid in real-world scenarios where a known, albeit suboptimal, safe strategy exists. In this paper, we adopt a similar assumption, as stated below.

**Assumption 2** *(Zero Starting Point Assumption): There exists an action $a^0 \in \mathcal{A}$ such that $\phi(a^0) = \mathbf{0} \in \mathbb{R}^d$.*

**Remark 1** *We highlight that for problems where the initial action is not at the origin and incurs a non-zero cost $\tau_0$, the original problem can be converted to an equivalent one that satisfies Assumption 2 through a simple translation. In the new problem, the safety threshold is adjusted to $\tau - \tau_0$.*

## 3 NON-CONVEX FEATURE SPACES

Non-convexity in feature space frequently arises in real-world bandit problems due to the inherent complexity or discrete nature of decision spaces, as well as feature transformations $\phi(.)$. Applications such as recommendation systems, financial management, and medical treatment selection often involve action sets that are neither convex nor star-convex, but rather composed of discrete or separated subsets. In this section, we define structures in the feature space commonly encountered in these applications. We begin with the following definition:

**Definition 1** *Let $\mathcal{F} \triangleq \{\phi(a) \in \mathbb{R}^d \mid a \in \mathcal{A}\}$.*

We begin by examining continuous non-convex sets with local properties centered around $\phi(a^0)$ and points near the optimal point $\phi(a^*)$ within the feature space. Beyond these localized properties, the set may take on any arbitrary form.

**Assumption 3 (*Local Point Assumption*)** *There exists $0 < \epsilon < \min\{L, \frac{\tau}{\sqrt{d}}\}$ such that for all $\mathbf{x} \in \mathcal{F}$, we have $\alpha \frac{x}{\|x\|} \in \mathcal{F}$ for some $\alpha \in [\epsilon, \frac{\tau}{\sqrt{d}}]$. Let $x^* = \phi(a^*)$ denote the optimal point. Then, either of the following conditions holds:*

1. *$\langle \phi(a^*), \gamma^* \rangle \leq \tau - \iota$, where $0 < \iota < L - \epsilon$, or*

2. *$\alpha x^* \in \mathcal{F}$ for all $\alpha \in [\frac{\tau}{\tau + \iota}, 1]$, with $0 < \iota$ such that $\iota \leq L - \epsilon \leq 1$.*

Note that Assumption 3 is not only rich enough to capture both star-convex and convex structures, but also applies to a wide range of non-convex and discrete real-world problems. In particular, when $\phi(.)$ is the identity mapping, these conditions apply directly to the action set $\mathcal{A}$.

**Why do we need the $\epsilon$- and $\iota$-neighborhood conditions?** Starting from the initial safe point, the agent must explore a small region around this point (the origin) to gather information about different directions. The $\iota$-neighborhood assumption ensures that the agent can explore a small area around the optimal point, particularly when the optimal point lies on the boundary of the constraint. Without this exploration, solving the problem would be impossible, as our lower bound in Theorem 2 demonstrates the necessity for $\epsilon$ and $\iota$ to be strictly positive.

**Real-World Implications of Assumption 3.** Consider an investment problem where a venture capitalist (VC) needs to decide how to allocate its funds. Suppose the VC can invest at different levels in a company and must determine how to hedge the associated risks. If the VC makes a small investment, it risks losing only a small amount of money if the company goes bankrupt. However, if the company does very well, the VC only owns a small portion, so the reward is also limited. The reverse is true if the VC makes a large investment. The VC could hedge its bets by initially making several small investments in different startups to gather information on how these investments perform. Once the VC identifies a promising startup, it can then take on more risk by making a larger investment in that company. In this context, the smaller, safer investments represent the $\epsilon$-condition in Assumption 3, meaning they are small and close to the origin (a safe point). On the other hand, the larger, higher-risk investments in the profitable startup corresponds to the $\iota$-condition, which is further from the origin, closer to the safety threshold, but with the potential for higher returns as its closer to the optimal decision(See Fig. 2b). For more example please see Appendix C

**Comparison of Assumption 3 and the Star-Convex Assumption in Pacchiano et al. (2024).** Assumption 3 is a local assumption, as it only imposes conditions on the neighborhoods around the starting and optimal points, with $\epsilon$ and $\iota$ being arbitrarily small. In contrast, star-convexity is a global assumption, requiring that all lines connecting any feature point to the starting point lie within the feature set $\mathcal{F}$. For a visual example, see Figs. 3a and 3b.

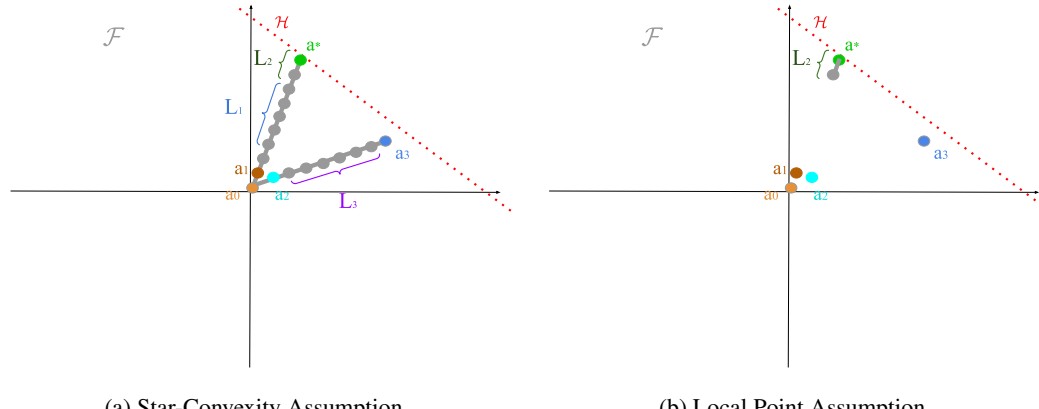

(a) Star-Convexity Assumption    (b) Local Point Assumption

Figure 3: (a) $\mathcal{H}$ denotes the constraint's boundry, where $a_0$ is the initial safe point (origin), and $a^*$ is the optimal point. Given that $a_0$, $a^*$, and $a_3$ are fixed in the feature space, star-convexity as described in Pacchiano et al. (2024) requires that the lines connecting $a_0$ to $a^*$ and $a_0$ to $a_3$ lie entirely within $\mathcal{F}$. (b) The Local Point Assumption only requires the points $a_1$, $a_2$, and the line segment $L_2$ to be in the feature space $\mathcal{F}$, without imposing the same requirement on the line segments $L_1$ and $L_3$.

## 4 OUR APPROACH

---

**Algorithm 1** Non-Convex Safe Linear UCB (NCS-LUCB)

---

**Require:** $\nu$, $\delta$, $\tau$, $\lambda$, $d$
1: **for** episode $t = 1, \ldots, T$ **do**
2:     $\Lambda_t = \Sigma_{\tau=1}^{t-1}\phi(a_\tau)\phi(a_\tau)^\top + \lambda I$
3:     $\theta_t = (\Lambda_t)^{-1}\Sigma_{\tau=1}^{t-1}\phi(a_\tau)r_\tau(a_\tau)$
4:     $\gamma_t = (\Lambda_t)^{-1}\Sigma_{\tau=1}^{t-1}\phi(a_\tau)c_\tau(a_\tau)$
5:     Calculate estimated safe set : $\mathcal{A}_t \triangleq \mathcal{A}_t^{\text{RLS}} \cup \mathcal{A}^{\frac{\tau}{\sqrt{d}}}$ according to Eqs. (1) and (2)
6:     Take action $a_t = argmax_{a \in \mathcal{A}_t}\langle\phi(a), \theta_t\rangle + b_t(a)$, where $b_t(.)$ defined in Eq.(3).
7:     Play $a_t$ and observe its reward $r_t$ and cost $c_t$.
8: **end for**

---

We introduce our algorithm, NCS-LUCB, as detailed in Algorithm 1. Inspired by the LC-LUCB algorithm from Pacchiano et al. (2024), our method significantly extends the approach to address non-convex and discrete problems. Our approach leverages UCB exploration while taking a conservative approach toward the costs associated with each action. A key innovation in our algorithm is a novel form of reward shaping, specifically designed to address the inherent non-convexity challenges within the feature space. Detailed explanations of the main steps are provided below.

**Reward and safe-set estimation.** We use Recursive Least Squares (RLS) to estimate the reward and safety parameters in lines $2 - 4$ of Algorithm 1. In line 5, we construct a conservative estimate of the safe set of actions based on both the RLS estimation and the Cauchy-Schwarz inequality. Specifically, $\mathcal{A}_t^{\text{RLS}}$ is defined as follows:

$$\mathcal{A}_t^{\text{RLS}} \triangleq \{a \in \mathcal{A} : \langle\phi(a), \gamma_t\rangle + \beta_2\|\phi(a)\|_{\Lambda_t^{-1}} \leq \tau\} \tag{1}$$

Theorem 2 from Abbasi-Yadkori et al. (2011) demonstrates that for any $\delta \in (0, 1)$, the choice of

$$\beta_2 = \sigma\sqrt{d\log\left(\frac{1+\frac{TL^2}{\lambda}}{\delta}\right)} + \sqrt{\lambda d}$$ ensures that $\mathcal{A}_t \subset \mathcal{A}^{\text{safe}}$ holds with probability $1 - \delta$. In addition

to $\mathcal{A}_t^{\text{RLS}}$, we also consider $\mathcal{A}^{\frac{\tau}{\sqrt{d}}}$ defined as:

$$\mathcal{A}^{\frac{\tau}{\sqrt{d}}} \triangleq \{a \in \mathcal{A} \mid \|\phi(a)\| \leq \frac{\tau}{\sqrt{d}}\} \tag{2}$$

Given Assumption 2 and the linearity of the problem (Assumption 1), the Cauchy-Schwarz inequality confirms that all actions in $\mathcal{A}^{\frac{\tau}{\sqrt{d}}}$ are safe, i.e., $\mathcal{A}^{\frac{\tau}{\sqrt{d}}} \subset \mathcal{A}^{\text{safe}}$.

**Bonus design.** At step 6, we implement an optimism-based approach to encourage the agent to select unexplored safe actions. The bonus expression is given by:

$$b_t(a) \triangleq \beta_1 \|\phi(a)\|_{(\Lambda_t)^{-1}} + g_t^\nu(a), \tag{3}$$

where the first term is a expression term used in uconstrained bandits literature, as discussed in Abbasi-Yadkori et al. (2011). However, in our case, since $a^*$ may not lie within the estimated safe set $\mathcal{A}_t$, we introduce a new bonus term, $g_t^\nu(.)$, to capture the distance between optimal point and the estimated safe set, defined as follows:

$$g_t^\nu(a) \triangleq \nu \times \left( 1 - \frac{\tau}{\tau + 2\beta_2 L \|\overline{\phi(a)}\|_{(\Lambda_t)^{-1}}} \right). \tag{4}$$

**Star-convex cases.** In star-convex case, setting $\nu = 1$ in Eq. 4 maintains the optimism property. This is because one can show that $\alpha \phi(a^*) \in \phi(\mathcal{A}_t)$ holds for some $\alpha \geq \frac{\tau}{\tau + 2\beta_2 L \|\phi(a^*)\|_{(\Lambda_t)^{-1}}}$.

Consequently, the distance between $\phi(a^*)$ and $\phi(\mathcal{A}_t)$ is less than $\left( 1 - \frac{\tau}{\tau + 2\beta_2 L \|\phi(a^*)\|_{(\Lambda_t)^{-1}}} \right)$.

However, when $\mathcal{F}$ is no longer star-convex, $\alpha \phi(a^*) \in \phi(\mathcal{A}_t)$ **does not** necessarily hold for all $0 \leq \alpha \leq \frac{\tau}{\tau + 2\beta_2 L \|\phi(a)\|_{(\Lambda_t)^{-1}}}$. In fact, setting $\nu = 1$ introduces a bias toward suboptimal directions, which we refer to as the **non-convexity bias**. This bias ultimately leads to linear regret. We elaborate on this bias in Section 5.2.

**Solving non-convexity bias.** When $\mathcal{F}$ is non-convex, $\phi(\mathcal{A}_t)$ also becomes non-convex, making the computation of the distance between $\phi(a^*)$ and $\phi(\mathcal{A}_t)$ intractable. However, to design an appropriate bonus term, calculating this distance is still necessary. To address this, in Lemma 2, we show that the distance between $\phi(a^*)$ and the features in the $\epsilon$-neighborhood of the origin acts as an upper bound for the distance between $\phi(a^*)$ and $\phi(\mathcal{A}_t)$ and can be used instead for bonus design. In this way, with an appropriate choice of $\nu$ as discussed in the lemma, the designed bonus term restores optimism, ensuring exploration in the optimal direction (see Appendix E).

*Note that, in addition to restoring optimism, it is crucial to ensure that $g_t^\nu(.)$ does not result in linear regret. Accordingly, in Lemma 4, we demonstrate that $g_t^\nu(.)$ converges to zero at an appropriate rate, resulting in a regret cost of $\tilde{\mathcal{O}}(d\frac{\sqrt{\tau}}{\epsilon\iota})$ in the upper bound.*

**Environment interaction.** In step 7, the algorithm plays the selected action, observes the reward and cost, and stores them for the next round. Steps 2–7 are repeated for $T$ rounds.

## 5 ANALYSIS

In this section, we present the main results of our study. We prove that Algorithm 1 achieves a sublinear regret. Also, we establish a lower bound that demonstrates the inherent impact of non-convexity on the performance of any near-optimal algorithm.

### 5.1 MAIN RESULTS

Our first result is a high-probability, sublinear upper bound on the performance of Algorithm 1.

**Theorem 1** *Consider a linear bandit problem under Assumptions 1, 2, and 3. In Algorithm 1, let $\nu = \frac{\tau + \iota}{\iota}$, $\beta_1 = \beta_2 = \sigma \sqrt{d \log \left( \frac{1 + \frac{TL^2}{\lambda}}{\delta} \right)} + \sqrt{\lambda d}$, and $\lambda = 1$. Then, for any $\delta \in (0, \frac{1}{2})$, with probability at least $1 - 2\delta$ Algortihm 1 remains safe, i.e., $\mathcal{A}_t \subset \mathcal{A}^{safe}$, $\forall t \in [T]$. Further, the regret of Algorithm 1 with probability at least $1 - 2\delta$ satisfies the following upper bound:*

$$Regret(T) \leq \left( 2\beta_1 + \frac{2\beta_2 L(\tau + \iota)}{\epsilon \iota \tau} \right) \sqrt{2Td \log(\frac{d\lambda + TL^2}{\lambda d})} \tag{5}$$

**Comparison with Theorem 18 in Pacchiano et al. (2024).** The key distinction between our upper bound and that presented in Pacchiano et al. (2024) is the coeffecicient $\frac{1}{\epsilon\iota}$, where $\epsilon$ and $\iota$ reflect our local Assumption 3 in a non-convex space. Additionally, in our work, $\tau$ represents the safety gap, assuming our algorithm starts from the origin at each state, resulting in an initial safe action with zero cost and a corresponding gap of $\tau$.

**Remark 2** *We highlight that the result of Theorem 1 naturally extends to linear contextual bandits. For further discussion, please refer to Appendix J.*

**Adapting to unknown $\iota$.** While the choice of $\nu$ does not affect safety in Theorem 1, selecting it appropriately is crucial for achieving sublinear regret. Theorem 1 defines $\nu$ in terms of $\iota$. However, when a meaningful lower bound for $\iota$ is unknown, the agent can adopt the Bandits over Bandits (BOB) approach, as proposed by Cheung et al. (2019). This method employs a two-layer meta-structure, where the base learner is NCS-LUCB, and the meta-learner adaptively selects $\iota$ based on the cumulative rewards of the base learner. The full implementation and analysis of this technique are beyond the scope of this work and are left for future research.

*To the best of our knowledge, this is the first such result in literature for non-convex linear bandits. Further in Theorem 2, we provide a minimax lower bound that verifies the role of $\iota$ and $\epsilon$ in the upper bound of Theorem 1.*

**Theorem 2** *(Lower bound of safe linear bandits with non-convex action space) Consider the setup defined in Theorem 1. Then, for all $\epsilon \in (0, \frac{1}{4})$, and $\iota \in (0, \frac{1}{4})$, and for all $T \geq \frac{32e}{\epsilon\iota^2}$, the following information-theoretic lower bound holds for any safe algorithm:*

$$Regret(T) \geq \max\{\frac{d}{8e^2}\sqrt{T}, \frac{1-2\epsilon}{\epsilon}(\frac{1-\iota}{\iota})^2\}. \tag{6}$$

**Remark 3** *Note that since $Regret(T) \leq \tilde{O}\left(\frac{\sqrt{T}}{\epsilon\iota}\right)$, for all $T = \lceil\frac{1}{\epsilon\iota^2}\rceil$, we have: $Regret(T) \leq \tilde{O}\left(\frac{1}{\epsilon^{3/2}\iota^2}\right)$, which shows only a $\frac{1}{\epsilon^{1/2}}$ gap between the lower and upper bounds.*

### 5.2 NON-CONVEXITY BIAS

As discussed in Section 4, the non-convex nature of $\mathcal{F}$ prevents the agent from taking the step size $\alpha = \frac{\tau}{\tau+2\beta_2\|\phi(a^*)\|_{\Lambda_t^{-1}}}\|\phi(a^*)\|$ toward the optimal point $\phi(a^*)$. In this section, we present a toy example demonstrating how a bonus design focused solely on safety limitations, without accounting for non-convexity, biases exploration towards suboptimal directions in a non-convex space. This prevents the agent from expanding the safe set $\mathcal{A}_t$ toward the optimal point and ultimately results in linear regret.

**Toy example on non-convexity bias.** Consider a non-convex safe linear bandit problem with the action set $\mathcal{A} = \left\{ a_0 = \begin{bmatrix} 0 \\ 0 \end{bmatrix}, a_1 = \begin{bmatrix} \frac{1}{3} \\ 0 \end{bmatrix}, a_2 = \begin{bmatrix} 0 \\ \frac{2}{3} \end{bmatrix}, a_3 = \begin{bmatrix} 1 \\ 0 \end{bmatrix} \right\}$, a safety threshold of $\tau = 0.95$, and a true reward vector $\theta^* = \begin{bmatrix} 1 \\ 1 \end{bmatrix}$. Additionally, let $\gamma^* = \begin{bmatrix} 0 \\ 0 \end{bmatrix}$ with the transformation $\phi(a) = a$, i.e., an identity transformation. With this setup, the entire action set $\mathcal{A}$ is safe, i.e., $\mathcal{A}^{\text{safe}} = \{a \in \mathcal{A} \mid \langle a, \gamma^* \rangle \leq \tau = 0.95\} = \mathcal{A}$, and the optimal action is $a_3$.

Now, suppose we design a safe algorithm where the agent knows $\theta^*$ but must estimate $\gamma^*$. Moreover, assume the agent knows that the maximum possible reward is less than 1, i.e., $r^* \leq 1$. At time $t = 1$, the agent estimates the safety factor as $\gamma_1 = \begin{bmatrix} 1 \\ 1 \end{bmatrix}$ and the safe set as $\mathcal{A}_t = \{a \in \mathcal{A} \mid \langle a, \gamma_t \rangle \leq \tau = 0.95\}$. As a result, the agent believes that points of the form $\begin{bmatrix} \alpha \\ 0 \end{bmatrix}$ or $\begin{bmatrix} 0 \\ \alpha \end{bmatrix}$, where $\alpha \leq 0.95$, are safe. Therefore, the estimated safe action set is $\mathcal{A}_t = \{a_0, a_1, a_2\}$.

We now demonstrate how neglecting non-convexity biases the agent toward $a_2$, a suboptimal direction. As mentioned, algorithms designed for star-convex problems, such as LC-LUCB in Pacchiano et al. (2024), focus solely on safety limitations in their bonus design. This leads the agent to consider

the distance from the safety region's boundary to the optimal point when designing the bonus. With the safety boundary along the $x$-axis set at $\max\{\alpha \leq 0.95\}$ and $r^* \leq 1$, the bonus for action $a_1$ on the $x$-axis is computed as $b_1(a_1) = 1 - 0.95 = \frac{1}{20}$. Similarly, for action $a_2$ on the $y$-axis, the bonus is $b_1(a_2) = \frac{1}{20}$. At time $t = 1$, the agent evaluates the actions as follows:

$$\langle a_1, \theta^* \rangle + b_t(a_1) = \frac{1}{3} + \frac{1}{20} \leq \frac{2}{3} = \langle \phi(a_2), \theta^* \rangle \leq \langle \phi(a_2), \theta^* \rangle + b_t(a_2), \tag{7}$$

where the last inequality holds because $b_t(.) \geq 0$. This inequality holds for all $t \in [T]$, as the bonus term $b_t$ is non-increasing over time in UCB-based methods. Hence, the agent always prefers $a_2$ over $a_1$ for all $t \in [T]$, based on the selection criterion $\arg\max_{a \in \mathcal{A}_t} \langle a, \theta^* \rangle + b_t(a)$. However, this bias toward $a_2$ is problematic. Since $a_2$ lies on the $y$-axis, the agent's estimated safe set $\mathcal{A}_t$ never expands in the $x$-direction. As a result, the optimal action $a_3$, which is on the $x$-axis, remains outside $\mathcal{A}_t$ for all $t$, leading to linear regret.

**What is the correct strategy?** An effective strategy is to account for both non-convexity and safety limitations in the bonus design. The bonus should be based on the distance from the optimal point to the nearest available action within the safety region, rather than to the boundary of the safety region. Therefore, in our toy problem, the bonus for action $a_1$ should be based on the distance between the optimal point and $a_1$, i.e., $b_t(a_1) = r^* - \frac{1}{3} = \frac{2}{3}$. This implies that the inequality in Eq.(7) no longer holds. As agent samples $a_2$ over time, the term $b_t(a_2)$ becomes smaller, which guarantees that the agent will play $a_1$ based on the selection criterion $\arg\max_{a \in \mathcal{A}_t} \langle a, \theta^* \rangle + b_t(a)$. *Note that, in practice, calculating the distance to a non-convex set of actions is challenging. However, by designing $g_t^\nu$ and setting $\nu = \frac{\tau + \iota}{\iota}$, as shown in Lemma 2, our bonus will be optimistic enough to address the bias problem while overcoming this computational challenge.*

### 5.3 Outline of Proof for Theorem 1

We outline the proof steps for Theorem 1 and explain how our bonus design addresses the non-convexity bias challenges mentioned in Section 5.2. For detailed proofs of Theorems 1 and 2, please refer to Appendix.

**Step 1.** Before proceeding, we introduce two important events:

**Definition 2** *The event $\mathcal{E}_1$ is defined as: $\mathcal{E}_1 \triangleq \{\mathcal{A}_t \subset \mathcal{A}^{safe} \ \forall t \in [T]\}$. Additionally, the event $\mathcal{E}_2$ is defined as $\mathcal{E}_2 \triangleq \{|\langle \phi(a), \theta_t \rangle - \langle \phi(a), \theta^* \rangle| \leq \beta_1 \|\phi(a)\|_{(\Lambda_t)^{-1}}, \ \forall(a, t) \in \mathcal{A} \times [T]\}$.*

Event $\mathcal{E}_1$ defines the set of outcomes where Algorithm 1 remains safe, while event $\mathcal{E}_2$ specifies the outcomes where the estimation of the reward function is sufficiently accurate. Naturally, our interest lies in scenarios where both events occur simultaneously. Lemma 1 shows that, under the settings outlined in Theorem 1, the intersection of these two events occurs with high probability (Abbasi-Yadkori et al., 2011).

**Lemma 1** *(Theorem 2 from Abbasi-Yadkori et al. (2011)) Under the setup of Theorem 1, and for any fixed $\delta \in (0, \frac{1}{2})$, the event $\mathcal{E} \triangleq \mathcal{E}_1 \cap \mathcal{E}_2$ holds with probability at least $1 - 2\delta$.*

The proof can be found in Appendix D. Next, we prove the optimism property conditioned on the event $\mathcal{E}$ under the bonus design in Algorithm 1.

**Lemma 2** *(Optimism): In algorithm 1, under the setup of Theorem 1, conditioned on the event $\mathcal{E}$, the inequality $\langle \phi(a^*), \theta^* \rangle \leq \max_{a \in \mathcal{A}_t} \langle \phi(a), \theta_t \rangle + b_t(a), \ \forall(t) \in [T]$ holds.*

The proof of Lemma 2 can be found in Appendix E.

**Step 2.** Our bonus design, along with Lemma 2, allows us to present the following decomposition that upper bounds the regret:

**Lemma 3** *Conditioned on the event $\mathcal{E}$, the regret of Algorithm 1 is upper bounded as follows:*

$$Regret(T) \leq \underbrace{2\beta_1 \Sigma_{t=1}^T \|\phi(a_t)\|_{(\Lambda_t)^{-1}}}_{\mathcal{T}_1} + \underbrace{\Sigma_{t=1}^T g_t^\nu(a_t)}_{\mathcal{T}_2}. \tag{8}$$

Note that $\mathcal{T}_2$ captures the effect of our new bonus term in Eq.(4). Term $\mathcal{T}_1$ appears in the unconstrained case, Abbasi-Yadkori et al. (2011), and we can bound this term in the same way. Next, we bound the term $\mathcal{T}_2$ as follows:

**Lemma 4** *Under the assumptions of Theorem 1, and conditioned on the event $\mathcal{E}$, the following holds:* $\mathcal{T}_2 = \Sigma_{t=1}^T g_t^\nu(a_t) \leq \frac{2\beta_2 L\nu}{\iota\epsilon\tau}\Sigma_{t=1}^T \|\phi(a_t)\|_{(\Lambda_t)^{-1}}$.

The proof of Lemma 4 can be found in Appendix G. Note that, the result of Lemma 4 enables us to utilize Lemma 11 in Abbasi-Yadkori et al. (2011) to show the sublinearity of $\mathcal{T}_2$.

**Step 3.** Now, by combining the last two steps and upper bounding the normalized term obtained in step 2, we can apply Lemma 11 from Abbasi-Yadkori et al. (2011) to obtain the final result.

## 6 NUMERICAL EXPERIMENTS

We conducted an experiment to evaluate the performance of the NCS-LUCB method and compare it with LC-LUCB from Pacchiano et al. (2024). [1] We considered a linear bandit scenario with a discrete action set $\mathcal{A} \triangleq = \{a_1, a_2, a_3, a_4, a_5\}$, where $a_1 = [0.1, 0]^\top$, $a_2 = [0, 0.4]^\top$, $a_3 = [0.8, 0]^\top$, $a_4 = [0, 0.7]^\top$, and $a_5 = [1, 0]^\top$. We set the parameters $\gamma^* = [1, 0]^\top$, $\theta^* = [1, 1]^\top$, $\delta = 0.01$, $\sigma = 0.01$, $\tau = 0.9$, $d = 2$, $T = 900000$, and chose the identity mapping for $\phi(.)$. In this setup, $a_5$ is considered unsafe, while the remaining actions are safe, i.e., $\mathcal{A}^{\text{safe}} = \{a_1, a_2, a_3, a_4\}$. Furthermore, $a_3$ is the optimal action, making $x$ the optimal direction and $y$ a suboptimal direction. For LC-LUCB, we set the bonus term as $b_t(a) := \alpha\beta\|\phi(a)\|_{(\Lambda_t)^{-1}}$, where $\alpha = (\frac{2}{\tau} + 2)$ and $\beta = \sigma\sqrt{d\log(\frac{1+T}{\delta})} + \sqrt{d}$, following Theorem 18 in Pacchiano et al. (2024). The regrets of NCS-LUCB and LC-LUCB from Pacchiano et al. (2024) are shown in Figs. 1b and 1a, respectively. Our algorithm achieves sublinear regret, indicating the successful expansion of the estimated safe set along the $x$-axis to include the optimal action $a_3$. In contrast, LC-LUCB converges to the suboptimal point $a_4$ and fails to expand its safe set along the $x$-axis to include the optimal point, resulting in linear regret. Additionally, the frequency of actions played by each algorithm is shown in Figs. 4a and 4b. As observed, our algorithm predominantly selects $a_1$ and $a_3$, which lie in the $x$-axis. However, LC-LUCB is biased toward the suboptimal $y$-axis, and mostly selects $a_4$. Notably, neither algorithm selects $a_5$, which is an unsafe action.

## 7 CONCLUSION

In this paper, we developed an algorithm to address the challenges of non-convex spaces in linear bandits with instantaneous hard constraints. Unlike previous works that relied on convexity and star-convexity, we demonstrated that local assumptions around the starting safe point and the optimal point are sufficient for near-optimal performance. We provided an upper bound that nearly matches the regret bounds under star-convexity and convexity assumptions, and a lower bound that highlights the necessity of parameters related to local assumptions in the upper bound. Additionally, our method also captures discrete cases with finite action spaces. An interesting area for future work is the adaptation of our approach to gradient-based methods. This adjustment is crucial, as the maximization step in non-convex scenarios often becomes intractable in non-convex continuous cases. Therefore, analyzing the impact of the non-convex optimization step on convergence is an essential next step.

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

**Organization of appendix** Appendix A includes figures supporting the numerical experiments discussed in Section 6. Appendix P presents an additional case study for simulation. Appendix B discusses related works. Appendix C provides additional real-world examples of Assumption 3. In Appendix D we provide the proof of Lemma 1. In Appendix E, we provide the proof for Lemma 2. We first state useful Lemmas 5 and 6 in Appendix E.1 where their proofs are provided in Appendix E.3. Then, proof of Lemma 2 is provided in Appendix E. Appendix F contains the proof for Lemma 3, while Appendix G provides the proof for Lemma 4. The proof of Theorem 1 is presented in Appendix H. Appendix I contains our proof for Theorem 2. Appendix J extends our work to linear contextual bandits. Exact same proof of linear bandits applies to the linear contextual bandits setup as well, however, for sake of completeness the complete proof for linear contextual bandits is provided in Appendices L M, N, O.

## A    SIMULATION FIGURES

This section provides supporting figures for the numerical experiments discussed in Section 6.

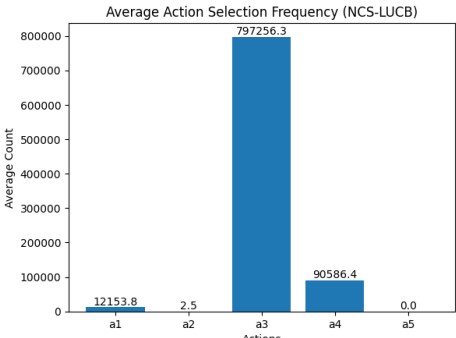
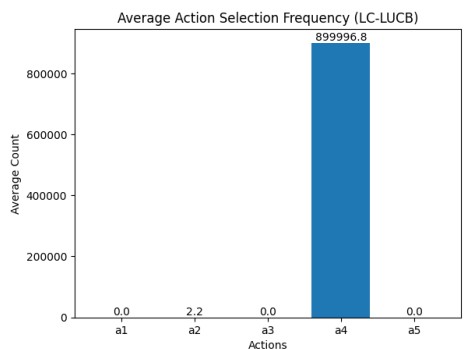

(a) Action selection frequency in NCS-LUCB (ours).

(b) Action selection frequency in LC-LUCB Pacchiano et al. (2024).

Figure 4: NCS-LUCB (ours) primarily explores the optimal direction along the $x$-axis by sampling from $a_1$ and $a_3$. In contrast, LC-LUCB from Pacchiano et al. (2024) is biased toward the suboptimal $y$-axis, predominantly sampling from $a_2$ and $a_4$. Each plot represents the average over 10 trials.

## B    RELATED WORKS

**RL with instantaneous hard constraints** Amani et al. (2021) solved the RL problem with instantenous constraint for linear MDP in star-convex spaces. Then, Amani & Yang (2022), solved safe problem for offline setup. Also, Wachi et al. (2021) solved the problem for generalized linear models. In all of these problems it is assumed that an initial safe action is known for each state, and safety is only related to unsafe actions (unsafe states does not exists). Then, Shi et al. (2023) studied problems with unsafe states in star-convex setting. Lastly, Wei et al. (2024) relaxed the assumption of a prior safe action is given to the algorithm but instead they can get sublinear constraint violation. Thus, none of the above works have studied RL with instantaneous hard constraints for non-convex feature spaces with local assumptions

**RL with cumulative constraints**: Lastly, RL problems with cumulative constraints are studied in Wu et al. (2016); Vaswani et al. (2022); Ghosh et al. (2022a); Ding & Lavaei (2023); Huang et al. (2023).

## C    APPLIED EXAMPLES OF ASSUMPTION 3

### C.1    SMART BUILDING MANAGEMENT

Consider a smart building management scenario where the goal is to schedule jobs while ensuring the total cost of the energy consumed remains below a set threshold. Each appliance can be operated

in either low-power or high-power mode. Running all appliances at full power would ensure jobs are completed quickly, but it risks causing an overload and exceeding the energy cost threshold. On the other hand, operating appliances in low-power mode will always keep the energy cost within the limit, but it is suboptimal in terms of job completion efficiency.

The manager might initially operate all appliances in low-power mode to gather information about the building's energy costs. Once sufficient information has been collected, the manager can selectively switch some appliances to high-power mode to improve job performance, while keeping others in low-power mode to ensure the total energy cost remains within the threshold. In this context, running appliances in low-power mode aligns with the $\epsilon$-condition in Assumption 3, where the system remains in a safer state and the energy cost is controlled. Conversely, operating some appliances at high power corresponds to the $\iota$-condition, which moves the system closer to the safety threshold but offers the potential for better job scheduling performance.

## C.2 Autonomous Vehicle Navigation: A Non-Convex Problem.

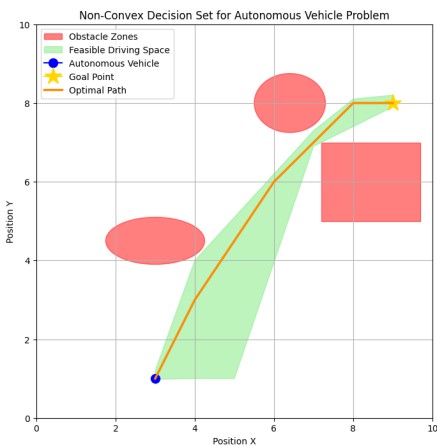

Figure 5: Autonomous vehicle with collision avoidance constraint is a non-convex problem.

## D Proof of Lemma 1

We can directly apply Theorem 2 from Abbasi-Yadkori et al. (2011) to show that each of the events $\mathcal{E}_1$ and $\mathcal{E}_2$ holds sepereatedly with a probability of at least $1 - \delta$. Using the union bound, we find that the event $\mathcal{E}_1 \cap \mathcal{E}_2$ holds with a probability of at least $1 - 2\delta$.

## E Optimism

### E.1 Preliminninary results

**Lemma 5** *Under Assumptions 1, 2, and 3, conditioned on the event $\mathcal{E}$, for all $t \in [T]$, we have $\alpha\phi(a^*) \in \phi(\mathcal{A}_t)$, where $\alpha \geq \epsilon$.*

The proof of this Lemma is provided in Appendix E.3.

**Lemma 6** *Let $\frac{\tau}{\tau+\iota} \leq \frac{\tau}{\tau+2\beta_2 L\|\phi(a^*)\|_{\Lambda_t^{-1}}}$. Then, under Assumptions 1- 3, we have $\alpha\phi(a^*) \in \phi(\mathcal{A}_t)$, for some $\alpha \in \left[\frac{\tau}{\tau+2\beta_2 L\|\phi(a^*)\|_{\Lambda_t^{-1}}}, 1\right]$.*

The proof of this Lemma is provided in Appendix E.3.

### E.2 PROOF OF LEMMA 2

Pick an arbitrary action $a' \in \mathcal{A}_t$. Then, we will have:

$$
\begin{aligned}
\max_{a\in\mathcal{A}_t}\langle\phi(a),\theta_t\rangle + b_t(a) &\geq \langle\phi(a'),\theta_t\rangle + b_t(a') \\
&= \langle\phi(a'),\theta_t\rangle + \beta_1\|\phi(a')\|_{\Lambda_t^{-1}} + g_t^\nu(a')
\end{aligned}
\tag{9}
$$

Now, on the event $\mathcal{E}_2$ we will have:

$$
\langle\phi(a'),\theta_t\rangle + \beta_1\|\phi(a')\|_{\Lambda_t^{-1}} + g_t^\nu(a') \geq \langle\phi(a'),\theta^*\rangle + g_t^\nu(a').
\tag{10}
$$

Thus, combining Equations 9 and 10 yields the following:

$$
\max_{a\in\mathcal{A}_t}\langle\phi(a),\theta_t\rangle + b_t(a) \geq \langle\phi(a'),\theta^*\rangle + g_t^\nu(a')
\tag{11}
$$

This brings us to analyze two sub-cases:

*Sub-case one:* Assume that $\frac{\tau}{\tau+2\beta_2 L\|\phi(a^*)\|_{\Lambda_t^{-1}}} \leq \frac{\tau}{\tau+\iota}$. Then, by Lemma 5, there exists an action $a_\alpha \in \mathcal{A}_t$ such that $\phi(a_\alpha) = \alpha\phi(a^*)$, where $\alpha \geq \epsilon$. Thus, by replacing $a'$ with $a_\alpha$ in Eq.(11) we have:

$$
\max_{a\in\mathcal{A}_t}\langle\phi(a),\theta_t\rangle + b_t(a) \geq \langle\phi(a_\alpha),\theta^*\rangle + g_t^\nu(a_\alpha) = \alpha\langle\phi(a^*),\theta^*\rangle + g_t^\nu(a_\alpha).
\tag{12}
$$

Now, using the definition of $g_t^\nu(.)$ in Eq.(4), and considering the fact that $\|\overline{\phi(a^*)}\|_{\Lambda_t^{-1}} = \|\overline{\alpha\phi(a^*)}\|_{\Lambda_t^{-1}}$, we have the following:

$$
g_t^\nu(a_\alpha) = \nu \times \left(1 - \frac{\tau}{\tau + 2\beta_2 L\|\overline{\phi(a^*)}\|_{(\Lambda_t)^{-1}}}\right)
\tag{13}
$$

By setting $\nu = \frac{\tau+\iota}{\iota}$, and considering that $\frac{\tau}{\tau+2\beta_2 L\|\phi(a^*)\|_{\Lambda_t^{-1}}} \leq \frac{\tau}{\tau+\iota}$ we obtain the following:

$$
g_t^\nu(a_\alpha) \geq \frac{(\tau+\iota)}{\iota}\left(\frac{\iota}{\tau+\iota}\right) = 1 \geq (1-\epsilon)
\tag{14}
$$

Since we assumed in our problem formulation that $r(a) \in [0,1]$ for all $a \in \mathcal{A}$, we obtain:

$$
g_t^\nu(a_\alpha) \geq (1-\epsilon) \geq (1-\epsilon)\langle\phi(a^*),\theta^*\rangle
\tag{15}
$$

Now, combining Equations (12) and (15) yields:

$$
\max_{a\in\mathcal{A}_t}\langle\phi(a),\theta_t\rangle + b_t(a) \geq \alpha\langle\phi(a^*),\theta^*\rangle + (1-\epsilon)\langle\phi(a^*),\theta^*\rangle \geq \langle\phi(a^*),\theta^*\rangle,
\tag{16}
$$

where the last inequality obtained by the fact that $\alpha \geq \epsilon$. This completes the proof for sub-case one.

*Sub-case two:* Now, assume that $\frac{\tau}{\tau+2\beta_2 L\|\overline{\phi(a^*)}\|_{\Lambda_t^{-1}}} \geq \frac{\tau}{\tau+\iota}$. Then, by Lemma 6, there exists an action $a_\alpha \in \mathcal{A}_t$ such that $\phi(a_\alpha) = \alpha\phi(a^*)$, where $\alpha \geq \frac{\tau}{\tau+2\beta_2 L\|\overline{\phi(a^*)}\|_{(\Lambda_t)^{-1}}}$. Thus, by replacing $a'$ with $a_\alpha$ in Eq.(11), we have:

$$\max_{a\in\mathcal{A}_t}\langle\phi(a),\theta_t\rangle + b_t(a) \geq \alpha\langle\phi(a^*),\theta^*\rangle + g_t^\nu(a_\alpha). \tag{17}$$

Now, similar to the sub-case one, by setting $\nu = \frac{\tau+\iota}{\iota} \geq 1$, we have:

$$\begin{aligned}
g_t^\nu(a_\alpha) =& \frac{\tau+\iota}{\iota} \times \left(1 - \frac{\tau}{\tau + 2\beta_2 L\|\overline{\phi(a^*)}\|_{(\Lambda_t)^{-1}}}\right) \\
\geq& \left(1 - \frac{\tau}{\tau + 2\beta_2 L\|\overline{\phi(a^*)}\|_{(\Lambda_t)^{-1}}}\right) \geq 1 - \alpha \geq (1-\alpha)\langle\phi(a^*),\theta^*\rangle
\end{aligned} \tag{18}$$

where the last inequality is obtained by the fact that $r(a) \in [0,1]$ for all $a \in \mathcal{A}$. Thus, combining Equations (17) and (18) yields:

$$\max_{a\in\mathcal{A}_t}\langle\phi(a),\theta_t\rangle + b_t(a) \geq \alpha\langle\phi(a^*),\theta^*\rangle + (1-\alpha)\langle\phi(a^*),\theta^*\rangle = \langle\phi(a^*),\theta^*\rangle. \tag{19}$$

This completes the proof for sub-case two as well as the Lemma 2.

### E.3 Proof of Lemmas 5 and 6

**Proof of Lemma 5:** By Assumption 3, there exists a positive number $\mu \in [\epsilon, \frac{\tau}{\sqrt{d}}]$ such that $\mu\frac{\phi(a^*)}{\|\phi(a^*)\|} \in \mathcal{F}$. Now, choose $\alpha = \frac{\mu}{\|\phi(a^*)\|}$. Then, we have the following:

$$\alpha\phi(a^*) = (\alpha\|\phi(a^*)\|) \times \frac{\phi(a^*)}{\|\phi(a^*)\|} = \mu\frac{\phi(a^*)}{\|\phi(a^*)\|},$$

which implies that $\alpha\phi(a^*) \in \mathcal{F}$. By Assumption 1, we have: $\|\phi(a^*)\| \leq L \leq 1$, which implies: $\alpha \geq \mu \geq \epsilon$. This implies that $\alpha\phi(a^*) \in \mathcal{F}$ for some $\alpha \geq \epsilon$. It remains to show that $\alpha\phi(a^*) \in \phi(\mathcal{A}_t)$ for all $t \in [T]$. Note that we can write the following:

$$\|\alpha\phi(a^*)\| = \|\mu\frac{\phi(a^*)}{\|\phi(a^*)\|}\| = \mu \leq \frac{\tau}{\sqrt{d}}, \tag{20}$$

which by Eq.(2) implies that $\alpha\phi(a^*) \in \phi(\mathcal{A}^{\frac{\tau}{\sqrt{d}}})$. Since $\mathcal{A}^{\frac{\tau}{\sqrt{d}}} \subset \mathcal{A}_t$, we have: $\alpha\phi(a^*) \in \phi(\mathcal{A}_t)$ as well, i.e., there exists an $a \in \mathcal{A}_t$ such that $\phi(a) = \alpha\phi(a^*)$, where $\alpha \geq \epsilon$. $\square$

**Proof of Lemma 6:** We decompose the proof of this lemma into two cases:

**Case 1:** Assume that $a^*$ does not lie on the constraint's boundary. Then, by Assumption 3 we have: $\langle\phi(a^*),\gamma^*\rangle \leq \tau - \iota$. Therefore, we can show that in this case, $a^* \in \mathcal{A}_t$. To prove our claim, note that, on the event $\mathcal{E}$, we have:

$$\begin{aligned}
\langle\phi(a^*),\gamma_t\rangle + \beta_2\|\phi(a^*)\|_{\Lambda_t^{-1}} &\leq \langle\phi(a^*),\gamma^*\rangle + 2\beta_2\|\phi(a^*)\|_{\Lambda_t^{-1}} \\
&= \tau - \iota + 2\beta_2\|\phi(a^*)\|_{\Lambda_t^{-1}}.
\end{aligned} \tag{21}$$

On the other hand, the condition $\frac{\tau}{\tau+\iota} \leq \frac{\tau}{\tau+2\beta_2 L\|\overline{\phi(a^*)}\|_{\Lambda_t^{-1}}}$ implies that $2\beta_2 L\|\overline{\phi(a^*)}\|_{\Lambda_t^{-1}} \leq \iota$. Thus, by Eq.(21), we have:

$$\langle\phi(a^*),\gamma_t\rangle + \beta_2\|\phi(a^*)\|_{\Lambda_t^{-1}} \leq \tau - \iota + 2\beta_2\|\phi(a^*)\|_{\Lambda_t^{-1}} \leq \tau$$

which implies $a^* \in \mathcal{A}_t^{\text{RLS}} \subset \mathcal{A}_t$. Thus, for $\alpha = 1$, we have $\alpha \phi(a^*) \in \phi(\mathcal{A}_t)$, which completes the proof for Case 1.

**Case 2:** Now, assume that $a^*$ lies on the constraint's boundry, i.e., we have $\langle \phi(a^*), \gamma^* \rangle = \tau$. We will show that $\alpha \phi(a^*) \in \phi(\mathcal{A}_t)$ for $\alpha = \frac{\tau}{\tau + 2\beta_2 L \|\overline{\phi(a^*)}\|_{\Lambda_t^{-1}}}$. To prove this, note that since $\alpha \geq \frac{\tau}{\tau + \iota}$, by Assumption 3, and because $a^*$ resides on the constraint's boundary, we must have $\alpha \phi(a^*) \in \phi(\mathcal{A})$, i.e., there exists an action $a_\alpha \in \mathcal{A}$ such that $\phi(a_\alpha) = \alpha \phi(a^*)$. Now, it remains to show that $a_\alpha \in \mathcal{A}_t$. Conditioned on the event $\mathcal{E}$, we have:

$$
\begin{aligned}
\langle \phi(a_\alpha), \gamma_t \rangle + \beta_2 \|\phi(a_\alpha)\|_{\Lambda_t^{-1}} &= \alpha \phi(a^*), \gamma_t \rangle + \beta_2 \|\phi(a^*)\|_{\Lambda_t^{-1}} \\
&\leq \alpha \left( \langle \phi(a^*), \gamma^* \rangle + 2\beta_2 \|\phi(a^*)\|_{\Lambda_t^{-1}} \right) \leq \alpha \left( \langle \phi(a^*), \gamma^* \rangle + 2\beta_2 L \|\overline{\phi(a^*)}\|_{\Lambda_t^{-1}} \right)
\end{aligned}
\tag{22}
$$

Now, by substituting $\langle \phi(a^*), \gamma^* \rangle = \tau$ and $\alpha = \frac{\tau}{\tau + 2\beta_2 L \|\overline{\phi(a^*)}\|_{\Lambda_t^{-1}}}$ in Eq. (22), we have:

$$
\langle \phi(a_\alpha), \gamma_t \rangle + \beta_2 \|\phi(a_\alpha)\|_{\Lambda_t^{-1}} \leq \frac{\tau}{\tau + 2\beta_2 L \|\overline{\phi(a^*)}\|_{\Lambda_t^{-1}}} \left( \tau + 2\beta_2 L \|\overline{\phi(a^*)}\|_{\Lambda_t^{-1}} \right) \leq \tau,
\tag{23}
$$

which implies $a_\alpha \in \mathcal{A}_t^{\text{RLS}} \subset \mathcal{A}_t$. This completes the proof of Case 2 and Lemma 6. $\square$

# F    PROOF OF LEMMA 3

**Proof of the Lemma 3:**    By utilizing Lemma 2, we can infer:

$$
\begin{aligned}
\text{Regret}(T) &= \Sigma_{t=1}^T \langle \phi(a^*), \theta^* \rangle - \langle \phi(a_t), \theta^* \rangle \leq \Sigma_{t=1}^T \langle \phi(a_t), \theta_t \rangle + b_t(a_t) - \langle \phi(a_t), \theta^* \rangle \\
&= \Sigma_{t=1}^T \langle \phi(a_t), \theta_t - \theta^* \rangle + \beta_1 \|\phi(a_t)\|_{\Lambda_t^{-1}} + g_t^\nu(a_t)
\end{aligned}
\tag{24}
$$

Now, conditioned on the event $\mathcal{E}$, we have $\langle \phi(a_t), \theta_t - \theta^* \rangle \leq \beta_1 \|\phi(a_t)\|_{\Lambda_t^{-1}}$. Thus, we can continue Eq.(24) as follows:

$$
\text{Regret}(T) \leq 2\beta_1 \Sigma_{t=1}^T \|\phi(a_t)\|_{\Lambda_t^{-1}} + \Sigma_{t=1}^T g_t^\nu(a_t)
\tag{25}
$$

This completes the proof. $\square$

# G    PROOF OF LEMMA 4

We begin by stating a Lemma that is helpful in the proof of Lemma 4.

**Lemma 7** *Under the setup of Theorem 1 and on the event $\mathcal{E}$, for every selected action $a_t$ in Algorithm 1, it holds that $\epsilon \leq \|\phi(a_t)\|, \forall t \in [T]$.*

The proof of this Lemma is provided in Appendix G.1. Now, we are ready for the proof of Lemma 4:

**Proof of Lemma 4:**    Using the expression for the $g_t^\nu(.)$ function, we have:

$$
\begin{aligned}
\Sigma_{t=1}^T g_t^\nu(a_t) &= \Sigma_{t=1}^T \nu \left( 1 - \frac{\tau}{\tau + 2\beta_2 L \|\overline{\phi(a_t)}\|_{(\Lambda_t)^{-1}}} \right) \\
&= (2\beta_2 L \nu) \Sigma_{t=1}^T \frac{\|\overline{\phi(a_t)}\|_{(\Lambda_t)^{-1}}}{\tau + 2\beta_2 L \|\overline{\phi(a_t)}\|_{(\Lambda_t)^{-1}}} \leq \frac{(2\beta_2 L \nu)}{\tau} \Sigma_{t=1}^T \|\overline{\phi(a_t)}\|_{(\Lambda_t)^{-1}},
\end{aligned}
\tag{26}
$$

where the last inequality follows from the fact that $\tau \leq \tau + 2\beta_2 L \|\overline{\phi(a_t)}\|_{(\Lambda_t)^{-1}}$. Now, since $\|\overline{\phi(a_t)}\|_{(\Lambda_t)^{-1}} = \|\frac{\phi(a_t)}{\|\phi(a_t)\|}\|_{(\Lambda_t)^{-1}}$, we can apply Lemma 7 as follows:

$$\Sigma_{t=1}^T g_t^\nu(a_t) \leq \frac{(2\beta_2 L\nu)}{\tau} \Sigma_{t=1}^T \|\overline{\phi(a_t)}\|_{(\Lambda_t)^{-1}} \leq \frac{(2\beta_2 L\nu)}{\epsilon\tau} \Sigma_{t=1}^T \|\phi(a_t)\|_{\Lambda_t^{-1}}, \tag{27}$$

which completes the proof. $\square$

### G.1 PROOF OF LEMMA 7

To prove Lemma 7, we utilize a contradiction strategy. Assume that for some $t \in [T]$, we have $\|\phi(a_t)\| < \epsilon$. Note that by Assumption 3, we have $\alpha \frac{\phi(a_t)}{\|\phi(a_t)\|} \in \phi(\mathcal{A})$, for some $\alpha \in [\epsilon, \frac{\tau}{\sqrt{d}}]$. Also, since $\alpha \leq \frac{\tau}{\sqrt{d}}$, we have: $\alpha \frac{\phi(a_t)}{\|\phi(a_t)\|} \in \phi(\mathcal{A}^{\frac{\tau}{\sqrt{d}}}) \subset \phi(\mathcal{A}_t)$, i.e., there exists an $a' \in \mathcal{A}_t$ such that $\phi(a') = \alpha \frac{\phi(a_t)}{\|\phi(a_t)\|}$. Now, we show that at time $t$, Algorithm 1 will prefer to choose $a'$ instead of $a_t$, which results in a contradiction and completes the proof. To show that the algorithm prefers to choose $a'$, we consider the following:

$$\langle \phi(a_t), \theta_t \rangle + b_t(a_t) = \langle \phi(a_t), \theta_t \rangle + \beta_1 \|\phi(a_t)\|_{\Lambda_t^{-1}} + g_t^\nu(a_t) \tag{28}$$

Now, conditioned on the event $\mathcal{E}$, we have $r(a_t) = \langle \phi(a_t), \theta^* \rangle \leq \langle \phi(a_t), \theta_t \rangle + \beta_1 \|\phi(a_t)\|_{\Lambda_t^{-1}}$. Since in our problem formulation we assume that $r(.) \in [0, 1]$, we can say the following:

$$0 \leq r(a_t) \leq \langle \phi(a_t), \theta_t \rangle + \beta_1 \|\phi(a_t)\|_{\Lambda_t^{-1}} \leq \frac{\epsilon}{\|\phi(a_t)\|} \left( \langle \phi(a_t), \theta_t \rangle + \beta_1 \|\phi(a_t)\|_{\Lambda_t^{-1}} \right), \tag{29}$$

where the last inequality follows from the fact that, according to the contradiction assumption, we have $\|\phi(a_t)\| \leq \epsilon$, which implies $1 \leq \frac{\epsilon}{\|\phi(a_t)\|}$. Now, combining Equations (28) and (29), we have:

$$\langle \phi(a_t), \theta_t \rangle + b_t(a_t) \leq \frac{\alpha}{\|\phi(a_t)\|} \left( \langle \phi(a_t), \theta_t \rangle + \beta_1 \|\phi(a_t)\|_{\Lambda_t^{-1}} \right) + g_t^\nu(a_t) \tag{30}$$

Now, substituting the definition of $\phi(a') = \alpha \frac{\phi(a_t)}{\|\phi(a_t)\|}$ into Eq. (30), we get:

$$\langle \phi(a_t), \theta_t \rangle + b_t(a_t) \leq \langle \phi(a'), \theta_t \rangle + \beta_1 \|\phi(a')\|_{\Lambda_t^{-1}} + g_t^\nu(a_t) \tag{31}$$

Now, using the definition of $g_t^\nu$ in Eq.(4), one can verify that $g_t^\nu(a') = g_t^\nu(a_t)$. Thus, we can continue Eq. 31 as follows:

$$\langle \phi(a_t), \theta_t \rangle + b_t(a_t) \leq \langle \phi(a'), \theta_t \rangle + b_t(a'). \tag{32}$$

The last inequality in Eq. (32) implies that in line 6 of Algorithm 1, action $a'$ is preferred to action $a_t$, which is contradiction and proves that $\|\phi(a_t)\| \geq \epsilon$, for all $t \in [T]$. $\square$

## H PROOF OF THEOREM 1

We apply Lemmas 3 and 4 to get the following:

$$\text{Regret}(T) \leq \Sigma_{t=1}^T (2\beta_1 + \frac{2\beta_2 L\nu}{\iota\epsilon\tau}) \|\phi(a_t)\|_{\Lambda_t^{-1}} \tag{33}$$

Following the steps outlined in the proof of Theorem 3 in Abbasi-Yadkori et al. (2011), we proceed as follows:

$$\Sigma_{t=1}^T \|\phi(a_t)\|_{\Lambda_t^{-1}} \le \sqrt{T\Sigma_{t=1}^T \|\phi(a_t)\|_{\Lambda_t^{-1}}^2} \tag{34}$$

By Assumption 1, given that $L \le 1$ and $\lambda = 1$ in Algorithm 1, it can be shown that $\|\phi(a_t)\|_{\Lambda_t^{-1}} \le 1$. Consequently, the following inequality holds:

$$\|\phi(a_t)\|_{\Lambda_t^{-1}}^2 \le 2\log(1+\|\phi(a_t)\|_{\Lambda_t^{-1}}^2). \tag{35}$$

Thus, by Equations 34 and 35 we have:

$$\sqrt{T\Sigma_{t=1}^T \|\phi(a_t)\|_{\Lambda_t^{-1}}^2} \le \sqrt{2T\Sigma_{t=1}^T \log(1+\|\phi(a_t)\|_{\Lambda_t^{-1}}^2)}$$
$$= \sqrt{2T(\log(\det(\Lambda_T)) - \log(\lambda^d))}. \tag{36}$$

where the last inequality is obtained by Lemma 11 from Abbasi-Yadkori et al. (2011). Now, considering that $\|\phi(a_t)\| \le L$, it follows that the trace of $\Lambda_T$ is bounded by $d\lambda + TL^2$. Since $\Lambda_T$ is a positive definite matrix, the determinant of $\Lambda_T$ can be bounded by:

$$det(\Lambda_T) \le (\frac{\text{trace}(\Lambda_T)}{d})^d \le (\frac{d\lambda + TL^2}{d})^d.$$

Combining everything together yields:

$$\sqrt{2T(\log(\det(\Lambda_T)) - \log(\lambda^d))} \le \sqrt{2T(\log((\frac{d\lambda + TL^2}{d})^d) - \log(\lambda^d))}$$
$$= \sqrt{2Td\log((\frac{d\lambda + TL^2}{\lambda d}))} = \sqrt{2Td\log((\frac{d\lambda + TL^2}{\lambda d}))} \tag{37}$$

Now, utilizing Equations (34) through (37), we conclude that:

$$\Sigma_{t=1}^T \|\phi(a_t)\|_{\Lambda_t^{-1}} \le \sqrt{2Td\log((\frac{d\lambda + TL^2}{\lambda d}))} \tag{38}$$

Now, by integrating Equations (33) and (38), we establish the desired upper bound as follows:

$$\text{Regret}(T) \le (2\beta_1 + \frac{2\beta_2 L\nu}{\epsilon\iota\tau})\sqrt{2Td\log((\frac{d\lambda + TL^2}{\lambda d}))}\square \tag{39}$$

# I  PROOF OF THEOREM 2

To establish the lower bound, we consider a scenario with a single state, analogous to a Linear Safe Bandit. Since the Bandit case is a subset of our RL problem, any lower bound derived for this scenario also applies to our problem as well.

## I.1  PRELIMINARY LEMMAS.

Our overall proof sketch is same as the Theorem 6 in Pacchiano et al. (2021) and Theorem 3 in Shi et al. (2023). Here, however, we extend the approach to accommodate a continuous and non-convex action space. We initiate with a divergence decomposition lemma adapted for continuous action spaces.

**Lemma 8** *(Divergence decomposition for more general action spaces (Lattimore & Szepesvári, 2020)). Consider a policy $\pi$, and let $\mathbb{P}_\mathcal{V} = \mathbb{P}_{\mathcal{V}\pi}$ and $\mathbb{P}_{\mathcal{V}'} = \mathbb{P}_{\mathcal{V}'\pi}$ denote the measures on the*

*canonical bandit model induced by $T$ round of interconnection of policy $\pi$ and environments $\mathcal{V}$ and $\mathcal{V}'$ respectively. Then the following holds:*

$$\mathbf{KL}(\mathbb{P}_\mathcal{V}, \mathbb{P}_{\mathcal{V}'}) = \mathbb{E}_\mathcal{V}[\Sigma_{t=1}^T \mathbf{KL}((P_{a^t}, Q_{a^t}), (P_{a'^t}, Q_{a'^t}))], \tag{40}$$

*where $P_{a^t}$ and $Q_{a^t}$ represent the reward and cost distributions corresponding to the action $a^t$ respectively.*

Since in bandits we do not have states (or only have one state), setting the feature function as an identity transformation makes the reward and cost linear functions of the actions, i.e., $r(a) = \langle a, \theta^* \rangle$ and $c(a) = \langle a, \gamma^* \rangle$ for all $a \in \mathcal{A}$. Assuming the noise in both reward and cost measurements is standard Gaussian, i.e., $\zeta^t, \eta^t \sim \mathcal{N}(0, 1)$, the distributions for the observed rewards and costs also become Gaussian:

$$\hat{r}(a^t) \sim \mathcal{N}(r(a^t), 1), \quad \hat{c}(a^t) \sim \mathcal{N}(c(a^t), 1), \quad \forall t \in [T].$$

Thus, the following Lemma facilitates the computation of divergence between two Gaussian distributions:

**Lemma 9** *(Divergence between two Gaussians (Lattimore & Szepesvári, 2020)). The divergence between two Gaussians distributions with means $\mu_1, \mu_2 \in \mathbb{R}$ and common variance $\sigma^2$ is given by:*

$$\mathbf{KL}(\mathcal{N}(\mu_1, \sigma^2), \mathcal{N}(\mu_2, \sigma^2)) = \frac{(\mu_1 - \mu_2)^2}{2\sigma^2} \tag{41}$$

Before delving into the proof of Theorem 2, we rewrite the useful Lemma 11 from Pacchiano et al. (2021). Define the binary relative entropy as follows:

$$d(x, y) \triangleq x \log(\frac{x}{y}) + (1 - x) \log(\frac{1-x}{1-y})$$

which for all $x \in [\frac{1}{2}, 1]$ and $y \in (0, 1)$ satisfies $d(x, y) \geq \frac{1}{2} \log(\frac{1}{4y})$. We then present the following lemma:

**Lemma 10** *(Lemma 11 in Pacchiano et al. (2021)) Consider the setup, definitions, and notations of the constrained bandits defined in Lemma 8. For a bandit environment, we define $\mathcal{F}_T$ as the filtration generated by the state-action sequences and the corresponding rewards and costs. Then, for any event $\mathcal{B}$ that is $\mathcal{F}_T$-measurable, the following holds:*

$$\mathbf{KL}(\mathbb{P}_\mathcal{V}, \mathbb{P}_{\mathcal{V}'}) \geq d(\mathbb{P}_\mathcal{V}(B), \mathbb{P}_{\mathcal{V}'}(B)) \tag{42}$$

### I.2    PROOF OF THEOREM 2

Note that by Theorem 24.1 from Lattimore & Szepesvári (2020), we know that $\text{Regret}(T) \geq \frac{d}{8e^2}\sqrt{T}$. Now it is remained to show that for $T \geq \frac{32e}{\epsilon \iota^2}$ we have $\text{Regret}(T) \geq \frac{1-2\epsilon}{\epsilon}(\frac{1-\iota}{\iota})^2$. First, we create two environments and demonstrate that for any policy, there exists an event $\mathcal{B}$ such that $d(\mathbb{P}_\mathcal{V}(\mathcal{B}), \mathbb{P}_{\mathcal{V}'}(\mathcal{B}))$ is lower bounded by a constant. Next, we establish a connection between the regret of the policy and the $\mathbf{KL}$ divergence of the two environments. Then, we apply Lemma 10 to derive our desired lower bound. Thus, we start by defining our environments.

**Environment description.**

**Action-Set:** We focus on the two-dimensional space where $a$, $\theta^*$, and $\gamma^*$ belong to $\mathbb{R}^2$. The action space, denoted as $\mathcal{A}$, is explicitly defined as the union of three distinct sub-spaces: $\mathcal{A} \triangleq \mathcal{A}_1 \cup \mathcal{A}_2 \cup \mathcal{A}_3$. These sub-spaces are defined as follows:

$$\begin{aligned} \mathcal{A}_1 &:= \{(x, y) \in \mathbb{R}^2 : |x| \leq \epsilon, |y| \leq \epsilon\} \\ \mathcal{A}_2 &:= \{(x, y) \in \mathbb{R}^2 : x = 0, \ 1 - \nu - \iota \leq y \leq 1 - \nu\} \\ \mathcal{A}_3 &:== \{(x, y) \in \mathbb{R}^2 : y = 0, \ -1 \leq x \leq -1 + \iota\}, \end{aligned} \tag{43}$$

where $\epsilon$ and $\iota$ are the parameters specified in Assumption 3. Additionally, we assume $L = 1$ and define $\nu$ as a sufficiently small positive parameter, ensuring that $\epsilon < 1 - \nu - \iota$ and $2\epsilon \leq 1 - \nu$. For the purpose of this proof, we set $\nu = \frac{1}{2}$, which implies $\epsilon \leq \frac{1}{4}$ and $\iota \leq \frac{1}{4}$.

Our main goal, as stated in Theorem 2, is to demonstrate that the dependence of the upper bound derived in Theorem 1 on the parameters $\iota$ and $\epsilon$ is essential. Specifically, if these parameters approach zero, the regret of any algorithm will tend towards infinity, at least for one environment adhering to the structure outlined in Theorem 1.

We are now prepared to describe two different environments, hereafter referred to as *Env1* and *Env2*.

**Env1:** The reward and cost parameters for *Env1* are defined as follows:

$$\theta_1^* = (-1, 1); \quad \gamma_1^* = (\frac{1}{1-\iota}, 0); \quad \tau = 1 \tag{44}$$

In this environment, the safety constraint $c(.) \leq \tau = 1$ implies that $\mathcal{A}_3$ is the unsafe set of actions. Also note that, the action $(0, 1 - \nu)$ emerges as the unique optimal action of *Env1*.

**Env2:** For the *Env2*, the reward and cost parameters are defined as follows:

$$\theta_2^* = \theta_1^* = (-1, 1); \quad \gamma_2^* = (1, 0); \quad \tau = 1 \tag{45}$$

The reward parameters of the two environments are identical; the primary distinction lies in the cost parameters. With $\gamma_2^*$, it is evident the entire set $\mathcal{A}$ is safe, with $(-1, 0)$ as the unique optimal point, which is located on the boundary of the constraint.

**Proof steps**   We aim to demonstrate that for all $T \geq \frac{32e}{\epsilon \iota^2}$, the regret of any algorithm must be at least $B \triangleq \frac{1-2\epsilon}{\epsilon}(\frac{1-\iota}{\iota})^2$. By contradiction, assume that there exists a safe-policy $\pi$ that achieves a regret lower than $B$ in both *Env1* and *Env2*, i.e., $B > \text{Regret}(T)$. Then, we define $\mathcal{B}$ as follows,

**Definition 3** *The $\mathcal{F}_T$-measurable set of events $\mathcal{B}$, is defined as follows:*

$$\mathcal{B} \triangleq \{\Sigma_{t=1}^T |a_y^t| > T\frac{(1-\nu-\epsilon)}{2}\}, \tag{46}$$

where $\{a^t\}_{t=1}^T$ represents the sequence of actions generated by the policy $\pi$ in *Env1* or *Env2*. Note that $a^t$ is a two-dimensional vector, where $a_x^t$ and $a_y^t$ are its corresponding $x$ and $y$ components, i.e., $a^t = (a_x^t, a_y^t)$.

Considering the definition of $\mathcal{B}$, we establish the following Lemma:

**Lemma 11** *In Env1, under the assumption that $B > Regret(T)$, the following inequality holds for all $T \in \mathbb{N}$:*

$$\mathbb{P}_1(\{\Sigma_{t=1}^T |a_y^t| > T\frac{(1-\nu-\epsilon)}{2}\}) \geq 1 - \frac{B}{T\frac{(1-\nu-\epsilon)}{2}} \tag{47}$$

**Proof:**   The proof of this Lemma can be found in Appendix I.3 .

Lemma 11 provides a critical insight into the behavior of algorithms operating in *Env1*. In fact, this Lemma establishes a lower bound on the cumulative absolute values of the $y$-components of the actions taken. This indicates that the algorithm consistently selects actions with significant large components along the $y$-axis. Such a pattern is consistent with the location of the optimal action on the $y$-axis. Consequently, any algorithm that achieves sublinear regret should mostly select actions from the set $\mathcal{A}_2$ in *Env1*.

In contrast, the optimal action in *Env2* lies along the $x$-axis, so we might expect a different pattern of action selection, as reflected in the following Lemma:

**Lemma 12** *In Env2, under the assumption that $B > Regret(T)$, the following inequality holds for all $T \in \mathbb{N}$:*

$$\mathbb{P}_2(\{\Sigma_{t=1}^T |a_y^t| > T\frac{(1-\nu-\epsilon)}{2}\}) \leq \frac{B}{\frac{\nu}{4}T} \tag{48}$$

**Proof:**   The detailed proof of this Lemma is available in Appendix I.3.

Note that at the beginning of the proof, we set $\nu = \frac{1}{2}$. According to Theorem 2, we select $T$ such that $T \geq \frac{32e}{\epsilon \iota^2}$. This choice of $T$ ensures that: $T \geq \max(\frac{B}{\frac{(1-\nu-\epsilon)}{4}}, \frac{16eB}{\nu})$. As a result, employing Lemmas 11 and 12 results in:

$$\mathbb{P}_1(\mathcal{B}) \geq \frac{1}{2}$$
$$\mathbb{P}_2(\mathcal{B}) \leq \frac{1}{4e} \tag{49}$$

Immediately after the inequalities obtained in Eq. (49), we can use Lemmas (8) to (10) to derive the following result:

**Lemma 13** *Under the assumption that $B > Regret(T)$, and noting that $L = 1$, for all $T \geq \frac{32e}{\epsilon \iota^2}$ the following inequality holds for the environment Env1:*

$$\mathbb{E}_1[\Sigma_{t=1}^T |a_x^t|] \geq \frac{(1-\iota)^2}{\iota^2} \tag{50}$$

**Proof:** The detailed proof of this lemma is provided in Appendix I.3.

This Lemma provides a lower bound on the cumulative absolute values of the $x$-components of actions taken in *Env1* by policy $\pi$. Notably, when $|a_x^t|$ is non-zero, it implies that $a^t$ belongs to $\mathcal{A}_1$ given that $\pi$ is a safe algorithm. This means $a^t$ represents a non-optimal action. Thus, we can utilize the lower bound in Eq. (50) to infer a lower bound for the regret of $\pi$ in *Env1*. Thus, consider the expression for the regret in *Env1*:

$$\begin{aligned}
Regret(T) &= \mathbb{E}_1[T(1-\nu) - \Sigma_{t=1}^T(-a_x^t + a_y^t)] \\
&= \mathbb{E}_1[T(1-\nu) - \Sigma_{t=1}^T(-a_x^t + a_y^t) \times (\mathbf{1}\{0 < |a_x^t|\} + \mathbf{1}\{|a_x^t| = 0\})] \\
&\geq \mathbb{E}_1[T(1-\nu) - \Sigma_{t=1}^T(-a_x^t + a_y^t) \times \mathbf{1}\{0 < |a_x^t|\}]
\end{aligned} \tag{51}$$

Now, for any safe algorithm in *Env1*, whenever $0 < |a_x^t|$ holds, we know that $a^t \in \mathcal{A}_1$. Thereby $a^t$ incures a regret of at least $1 - \nu - 2\epsilon$. Consequently:

$$\begin{aligned}
Regret(T) &\geq \mathbb{E}_1[T(1-\nu) - \Sigma_{t=1}^T(-a_x^t + a_y^t) \times \mathbf{1}\{0 < |a_x^t|\}] \\
&\geq \mathbb{E}_1[(1-\nu-2\epsilon)\Sigma_{t=1}^T \mathbf{1}\{0 < |a_x^t|\}] = (1-\nu-2\epsilon)\mathbb{E}_1[\Sigma_{t=1}^T \mathbf{1}\{0 < |a_x^t|\}]
\end{aligned} \tag{52}$$

Whenever $a^t \in \mathcal{A}_1$, we have $|a_x^t| \leq \epsilon$ which implies $\frac{|a_x^t|}{\epsilon} \leq \mathbf{1}\{0 < |a_x^t|\}$. Thus, we can continue from Eq. (52) as follows:

$$\begin{aligned}
B > Regret(T) &\geq (1-\nu-2\epsilon)\mathbb{E}_1[\Sigma_{t=1}^T \mathbf{1}\{0 < |a_x^t|\}] \\
&\geq (1-\nu-2\epsilon)\mathbb{E}_1[\Sigma_{t=1}^T \frac{|a_x^t|}{\epsilon}] \geq \frac{(1-\nu-2\epsilon)}{\epsilon} \times \frac{(1-\iota)^2}{\iota^2}
\end{aligned} \tag{53}$$

where the last inequality leverges Lemma 13.

Substituting $\nu = \frac{1}{2}$ into the last inequality, we obtain:

$$B > \frac{(\frac{1}{2} - 2\epsilon)}{\epsilon} \times \frac{(1-\iota)^2}{\iota^2} = B \tag{54}$$

which results in a contradiction, as $B$ cannot be strictly larger than itself. Thus, proof is complete. $\square$

### I.3 PROOFS OF LEMMAS 11, 12, 13.

**Proof of Lemma 11:** Under the assumption that $B > Regret(T)$, we can write the following inequalities:

$$B > Regret(T) = \mathbb{E}_1[T(1-\nu) - \Sigma_{t=1}^T\langle\theta_1^*, a^t\rangle] = \mathbb{E}_1[T(1-\nu) - \Sigma_{t=1}^T(-a_x^t + a_y^t)], \tag{55}$$

where $\mathbb{E}_1$ is the expectation regarding the measure induced by policy $\pi$ and the environment's dynamic in *Env1* . Now we can continue the Eq. (55) as follows:

$$
\begin{aligned}
B &> \mathbb{E}_1[T(1-\nu) - \Sigma_{t=1}^T(-a_x^t + a_y^t)] \\
&= \mathbb{E}_1[(T(1-\nu) - \Sigma_{t=1}^T(-a_x^t + a_y^t)) \\
&\quad \times (\mathbf{1}\{\Sigma_{t=1}^T|a_y^t| > T\frac{(1-\nu)}{2}\} + \mathbf{1}\{\Sigma_{t=1}^T|a_y^t| \le T\frac{(1-\nu)}{2}\})],
\end{aligned}
\tag{56}
$$

where $\mathbf{1}\{.\}$ is an indicator function. Now, note that since the maximum reward that is achievable by a safe action in *Env1* is $1 - \nu$, then we know that $T(1-\nu) - \Sigma_{t=1}^T(-a_x^t + a_y^t) \ge 0$. As a result we can continue the Eq. (56) as follows:

$$
\begin{aligned}
B &> \mathbb{E}_1[(T(1-\nu) - \Sigma_{t=1}^T(-a_x^t + a_y^t)) \\
&\quad \times (\mathbf{1}\{\Sigma_{t=1}^T|a_y^t| > T\frac{(1-\nu)}{2}\} + \mathbf{1}\{\Sigma_{t=1}^T|a_y^t| \le T\frac{(1-\nu)}{2}\})] \\
&\ge \mathbb{E}_1[(T(1-\nu) - \Sigma_{t=1}^T(-a_x^t + a_y^t)) \times \mathbf{1}\{\Sigma_{t=1}^T|a_y^t| \le T\frac{(1-\nu)}{2}\}].
\end{aligned}
\tag{57}
$$

But note that since $-\epsilon \le a_x^t$, one can extend the last inequality as follows:

$$
\begin{aligned}
B &> \mathbb{E}_1[(T(1-\nu-\epsilon) - \Sigma_{t=1}^T a_y^t) \times \mathbf{1}\{\Sigma_{t=1}^T|a_y^t| \le T\frac{(1-\nu)}{2}\}] \\
&\ge \mathbb{E}_1[(T(1-\nu-\epsilon) - \Sigma_{t=1}^T a_y^t) \times \mathbf{1}\{\Sigma_{t=1}^T|a_y^t| \le T\frac{(1-\nu-\epsilon)}{2}\}],
\end{aligned}
\tag{58}
$$

where the last inequality is obtained by the fact that $\mathbf{1}\{\Sigma_{t=1}^T|a_y^t| \le T\frac{(1-\nu-\epsilon)}{2}\} \le \mathbf{1}\{\Sigma_{t=1}^T|a_y^t| \le T\frac{(1-\nu)}{2}\}$. Now, we can continue as follows:

$$
\begin{aligned}
B &> \mathbb{E}_1[(T(1-\nu-\epsilon) - \Sigma_{t=1}^T a_y^t) \times \mathbf{1}\{\Sigma_{t=1}^T|a_y^t| \le T\frac{(1-\nu-\epsilon)}{2}\}] \\
&= \mathbb{E}_1[(T(1-\nu-\epsilon) - T\frac{(1-\nu-\epsilon)}{2}) \times \mathbf{1}\{\Sigma_{t=1}^T|a_y^t| \le T\frac{(1-\nu-\epsilon)}{2}\}] \\
&= T\frac{(1-\nu-\epsilon)}{2}\mathbb{E}_1[\mathbf{1}\{\Sigma_{t=1}^T|a_y^t| \le T\frac{(1-\nu-\epsilon)}{2}\}] \\
&= T\frac{(1-\nu-\epsilon)}{2}\mathbb{P}_1(\Sigma_{t=1}^T|a_y^t| \le T\frac{(1-\nu-\epsilon)}{2})
\end{aligned}
\tag{59}
$$

Now by applying the complement rule of a probability measure,

$$
\begin{aligned}
\mathbb{P}_1(\{\Sigma_{t=1}^T|a_y^t| > T\frac{(1-\nu-\epsilon)}{2}\}) &= 1 - \mathbb{P}_1(\{\Sigma_{t=1}^T|a_y^t| \le T\frac{(1-\nu-\epsilon)}{2}\}) \\
&\ge 1 - \frac{B}{T\frac{(1-\nu-\epsilon)}{2}}
\end{aligned}
\tag{60}
$$

**Proof of Lemma 12:**  Now, similar to the previous step, under the assumption that $B \ge \text{Regret}(T)$ we will have:

$$
B > \text{Regret}(T) = \mathbb{E}_2[T - \Sigma_{t=1}^T\langle\theta_2^*, a^t\rangle] = \mathbb{E}_2[T - \Sigma_{t=1}^T(-a_x^t + a_y^t)],
\tag{61}
$$

where $\{a^t\}_{t=1}^T$ is the sequence of actions generated by the policy $\pi$ in *Env2*, and $\mathbb{E}_2$ is the expectation regarding the measure induced by policy and environment's dynamic in  *Env2*.

Now, same as what we have done for *Env1*, we can continue the Eq. (61) as follows:

$$
\begin{aligned}
B &> \mathbb{E}_2[T - \Sigma_{t=1}^T(-a_x^t + a_y^t)] \\
&\ge \mathbb{E}_2[(T - \Sigma_{t=1}^T(-a_x^t + a_y^t)) \times \mathbf{1}\{\Sigma_{t=1}^T|a_y^t| > T\frac{(1-\nu-\epsilon)}{2}\}]
\end{aligned}
\tag{62}
$$

In *Env2*, whenever the event $\{\Sigma_{t=1}^T|a_y^t| > T\frac{(1-\nu-\epsilon)}{2}\}$ occurs, it implies that at least a quarter of the samples are taken from $\mathcal{A}_1 \cup \mathcal{A}_2$. To establish this, we employ a contradiction strategy. We assume

that the fraction of samples taken from the region $\mathcal{A}_1 \cup \mathcal{A}_2$ is denoted by $r$, and that $0 \leq r < \frac{1}{4}$. Noting that all samples from $\mathcal{A}_3$ have a zero $y$-component ($a_y^t = 0$). Thus, we have:

$$\Sigma_{t=1}^T |a_y^t| < \frac{1}{4}(1 - \nu)T \tag{63}$$

Given that $2\epsilon \leq 1 - \nu$ we will have:

$$\begin{aligned} \Sigma_{t=1}^T |a_y^t| &< \frac{1}{4}(1 - \nu)T = \frac{1}{2}(1 - \nu)T - \frac{1}{4}(1 - \nu)T \leq \frac{1}{2}(1 - \nu)T - \frac{1}{2}\epsilon T \\ &= T\frac{(1 - \nu - \epsilon)}{2} \end{aligned} \tag{64}$$

where the last inequality results in a contradiction, affirming our arguement.

Now, since at least $\frac{1}{4}$ of samples are taken from $\mathcal{A}_1 \cup \mathcal{A}_2$, and considering that the action yielding the highest reward in this region ($\mathcal{A}_1 \cup \mathcal{A}_2$) is $(0, 1 - \nu)$, we can continue the inequality 62 as follows:

$$\begin{aligned} B &> \mathbb{E}_2[(T - \Sigma_{t=1}^T(-a_x^t + a_y^t)) \times \mathbf{1}\{\Sigma_{t=1}^T |a_y^t| > T\frac{(1 - \nu - \epsilon)}{2}\}] \\ &\geq \mathbb{E}_2[(\frac{\nu}{4}T) \times \mathbf{1}\{\Sigma_{t=1}^T |a_y^t| > T\frac{(1 - \nu - \epsilon)}{2}\}] \\ &= \frac{\nu}{4}T\mathbb{E}_2[\mathbf{1}\{\Sigma_{t=1}^T |a_y^t| > T\frac{(1 - \nu - \epsilon)}{2}\}] = \frac{\nu}{4}T\mathbb{P}_2(\{\Sigma_{t=1}^T |a_y^t| > T\frac{(1 - \nu - \epsilon)}{2}\}) \end{aligned} \tag{65}$$

As a result,

$$\mathbb{P}_2(\{\Sigma_{t=1}^T |a_y^t| > T\frac{(1 - \nu - \epsilon)}{2}\}) \leq \frac{B}{\frac{\nu}{4}T}\square \tag{66}$$

**Proof of Lemma 13** By Lemmas 8 -10 and using the inequalities obtained in Eq. (49) we will have:

$$\begin{aligned} &\mathbf{KL}(\mathbb{P}_1, \mathbb{P}_2) \\ &= E_1[\Sigma_{t=1}^T \mathbf{KL}(\mathcal{N}([\langle a^t, \theta_1^* \rangle, \langle a^t, \gamma_1^* \rangle]^T, \mathbf{I}_2), \ \mathcal{N}([\langle a^t, \theta_2^* \rangle, \langle a^t, \gamma_2^* \rangle]^T, \mathbf{I}_2)] \\ &= \frac{\iota^2}{2(1 - \iota)^2}\mathbb{E}_1[\Sigma_{t=1}^T |a_x^t|^2] \geq d(\mathbb{P}_1(\mathcal{B}), \mathbb{P}_2(\mathcal{B})) \geq \frac{1}{2} \end{aligned} \tag{67}$$

where the last inequality is obtained by inequalities obtained in Equations (49) and (49).

Now, since $L = 1$, we know that $|a_x^t| \leq 1$ which yields $|a_x^t|^2 \leq |a_x^t|$. Combining all together and using the Eq. (67),

$$\mathbb{E}_1[\Sigma_{t=1}^T |a_x^t|] \geq \frac{2}{2}\mathbb{E}_1[\Sigma_{t=1}^T |a_x^t|^2] \geq d(\mathbb{P}_1(\mathcal{O}), \mathbb{P}_2(\mathcal{O})) \geq \frac{(1 - \iota)^2}{\iota^2}\square \tag{68}$$

## J  LINEAR CONTEXTUAL BANDITS

Linear contextual bandits can be considered a special case of linear MDP, when the horizon $H = 1$, and there is no transition (Agrawal & Devanur, 2016; Zhu et al., 2023; Ghosh et al., 2022b; Amani et al., 2021). Therfore, we denote the linear contextual bandits as $(\mathcal{S}, \mathcal{A}, r, c)$, where $\mathcal{S}$ denotes the set of context. Then we will have the following Assumption:

**Assumption 4** *(Linear bandits Amani et al. (2019), Pacchiano et al. (2024)) Consider a constrained contextual bandits denoted as $(\mathcal{S}, \mathcal{A}, r, c)$, which is assumed to be a linear contextual bandits with a feature function $\phi : \mathcal{S} \times \mathcal{A} \rightarrow \mathcal{F} \subset \mathbb{R}^d$. Specifically, there exist unknown vectors $\theta^*, \gamma^* \in \mathbb{R}^d$ such that for any pair $(s, a) \in \mathcal{S} \times \mathcal{A}$, the cost function, and reward function are given by: $r(s, a) = \langle \phi(s, a), \theta^* \rangle$, and $c(s, a) = \langle \phi(s, a), \gamma^* \rangle$, respectively. Additionally, we assume without loss of generality that for all $(s, a) \in \mathcal{S} \times \mathcal{A}$, we have $\|\phi(s, a)\| \leq L$ for some $L \in (0, 1]$, and $\max(\|\theta^*\|, \|\gamma^*\|) \leq \sqrt{d}$, where $d$ is the dimension of the feature space.*

Similar to linear bandits, we assume that the agent has access to least one kown safe action at the begining of the algorithm. Therefore, we adapt the Assumption 2 for linear contextual bandits as follows

**Assumption 5** *(Zero starting point in in linear contextual bandits):* *For each $s \in \mathcal{S}$, there exists an action $a_s^0 \in \mathcal{A}$ such that $\phi(s, a_s^0) = \mathbf{0} \in \mathbb{R}^d$.*

Since our focus is on non-convex spaces, we need to adapt Assumption 3 to the linear contextual bandit setting. We begin with the following definition:

**Definition 4** *For each $s \in \mathcal{S}$, tet $\mathcal{F}_s \triangleq \{\phi(s, a) \in \mathbb{R}^d \mid a \in \mathcal{A}\}$.*

We are now ready to present our non-convex assumption for linear contextual bandits:

**Assumption 6** *(Local point assumption in linear contextual bandits )* *There exists $0 < \epsilon < \min\{L, \frac{\tau}{\sqrt{d}}\}$ such that for all $s$, and $\mathbf{x} \in \mathcal{F}_s$, we have $\alpha \frac{x}{\|x\|} \in \mathcal{F}_s$ for some $\alpha \in [\epsilon, \frac{\tau}{\sqrt{d}}]$. Let $x_s^* = \phi(s, a_s^*)$ denote the optimal point given the context $s$. Then, either of the following conditions holds:*

1. *$\langle \phi(s, a_s^*), \gamma^* \rangle \leq \tau - \iota$, where $0 < \iota < L - \epsilon$, or*

2. *$\alpha x_s^* \in \mathcal{F}_s$ for all $\alpha \in [\frac{\tau}{\tau + \iota}, 1]$, with $0 < \iota$ such that $\iota \leq L - \epsilon \leq 1$.*

We now present the following algorithm for non-convex linear contextual bandit settings with instantaneous hard constraints:

---
**Algorithm 2** Non-Convex Safe Linear Contextual UCB (NCSC-LUCB)

---
**Require:** $\nu$, $\delta$, $\tau$, $\lambda$, $d$
1: **for** episode $t = 1, \ldots, T$ **do**
2: $\quad \Lambda_t = \Sigma_{\tau=1}^{t-1} \phi(s_\tau, a_\tau) \phi(s_\tau, a_\tau)^\top + \lambda I$
3: $\quad \theta_t = (\Lambda_t)^{-1} \Sigma_{\tau=1}^{t-1} \phi(s_\tau, a_\tau) r_\tau(s_\tau, a_\tau)$
4: $\quad \gamma_t = (\Lambda_t)^{-1} \Sigma_{\tau=1}^{t-1} \phi(s_\tau, a_\tau) c_\tau(s_\tau, a_\tau)$
5: $\quad$ For each $s \in \mathcal{S} : \mathcal{A}_t(s) \triangleq \mathcal{A}_t^{\text{RLS}}(s) \cup \mathcal{A}^{\frac{\tau}{\sqrt{d}}}(s)$ according to Eq. (69)
6: $\quad$ Take action $a_t = argmax_{a \in \mathcal{A}_t(s)} \langle \phi(s, a), \theta_t \rangle + b_t(s, a)$, where $b_t(.)$ defined in Eq.(70).
7: $\quad$ Play $a_t$ and observe its reward $r_t$ and cost $c_t$.
8: **end for**

---

where $\mathcal{A}_t^{\text{RLS}}(s)$ , $\mathcal{A}^{\frac{\tau}{\sqrt{d}}}(s)$ are defined as follows:

$$\mathcal{A}_t^{\text{RLS}}(s) \triangleq \{a \in \mathcal{A} : \langle \phi(s, a), \gamma_t \rangle + \beta_2 \|\phi(s, a)\|_{\Lambda_t^{-1}} \leq \tau\}.$$
$$\mathcal{A}^{\frac{\tau}{\sqrt{d}}}(s) \triangleq \{a \in \mathcal{A} \mid \|\phi(s, a)\| \leq \frac{\tau}{\sqrt{d}}\}. \tag{69}$$

Also, the bonus $b_t(.)$ is defined as :

$$b_t(s, a) \triangleq \beta_1 \|\phi(s, a)\|_{(\Lambda_t)^{-1}} + g_t^\nu(s, a), \tag{70}$$

where $g_t^\nu(.)$, defined as follows:

$$g_t^\nu(s, a) \triangleq \nu \times \left(1 - \frac{\tau}{\tau + 2\beta_2 L \|\overline{\phi(s, a)}\|_{(\Lambda_t)^{-1}}}\right). \tag{71}$$

Now, we are ready to state our result for linear contextual bandits:

**Theorem 3** *Consider a linear contextual bandit under Assumptions 4, 5, and 6. In Algorithm 2, let*

$\nu = \frac{\tau + \iota}{\iota}$, $\beta_1 = \beta_2 = \sigma\sqrt{d \log\left(\frac{1 + \frac{TL^2}{\lambda}}{\delta}\right)} + \sqrt{\lambda d}$, *and* $\lambda = 1$. *Then, for any* $\delta \in (0, \frac{1}{2})$, *with the*

*probability of at least* $1 - 2\delta$ *Algortihm 2 remains safe, i.e.,* $\mathcal{A}_t(s_t) \subset \mathcal{A}^{safe}$, $\quad \forall t \in [T]$. *Further, the regret of Algorithm 2 with a probability of at least* $1 - 2\delta$ *satisfies the the following upper bound:*

$$Regret(T) \leq (2\beta_1 + \frac{2\beta_2 L\nu}{\epsilon\iota\tau})\sqrt{2Td\log(\frac{d\lambda + TL^2}{\lambda d})}. \tag{72}$$

The proof steps of Theorem 3 follow the same structure as those of Theorem 1, but we provide them here for completeness.

**Step 1.** Let $\mathcal{A}^{safe}(s) \triangleq \{a \in \mathcal{A} \mid \langle \phi(s,a), \gamma^* \rangle \leq \tau\}$. Then, we introduce two important events:

**Definition 5** *The event* $\mathcal{E}_1^C$ *is defined as:* $\mathcal{E}_1^C \triangleq \{\mathcal{A}_t(s) \subset \mathcal{A}^{safe}(s) \ \forall(s,t) \in \mathcal{S} \times [T]\}$. *Additionally, the event* $\mathcal{E}_2^C$ *is defined as* $\mathcal{E}_2^C \triangleq \{|\langle\phi(s,a), \theta_t\rangle - \langle\phi(s,a), \theta^*\rangle| \leq \beta_1 \|\phi(s,a)\|_{(\Lambda_t)^{-1}}, \ \forall(s,a,t) \in \mathcal{S} \times \mathcal{A} \times [T]\}$.

Then, the following Lemma shows that event $\mathcal{E}_1^C \cap \mathcal{E}_2^C$ holds with a high probability:

**Lemma 14** *(Theorem 2 from Abbasi-Yadkori et al. (2011)) Under the setup of Theorem 3 and for any fixed* $\delta \in (0, \frac{1}{2})$, *the event* $\mathcal{E}^C \triangleq \mathcal{E}_1^C \cap \mathcal{E}_2^C$ *holds with probability at least* $1 - 2\delta$.

The proof can be found in Appendix K.

Now, using Lemma 14, we can prove the optimism property as stated in the following Lemma:

**Lemma 15** *(Optimism): In algorithm 2, under the setup of Theorem 3, conditioned on the event* $\mathcal{E}^C$, *the inequality* $\langle\phi(s,a^*), \theta^*\rangle \leq \max_{a \in \mathcal{A}_t(s)}\langle\phi(s,a), \theta_t\rangle + b_t(s,a), \ \forall(s,t) \in \mathcal{S} \times [T]$ *holds.*

**Step 2.** Our bonus design, along with Lemma 15, allows us to present the following decomposition that upper bounds the regret:

**Lemma 16** *Conditioned on the event* $\mathcal{E}^C$, *the regret of Algorithm 2 is upper bounded as follows:*

$$Regret(T) \leq \underbrace{2\beta_1\Sigma_{t=1}^T\|\phi(s_t,a_t)\|_{(\Lambda_t)^{-1}}}_{\mathcal{T}_1} + \underbrace{\Sigma_{t=1}^T g_t^\nu(s_t,a_t)}_{\mathcal{T}_2}. \tag{73}$$

Terms $\mathcal{T}_1$ appears in the unconstrained case, Abbasi-Yadkori et al. (2011), and we can bound it in the same way. Next, we bound the term $\mathcal{T}_2$ as follows:

**Lemma 17** *Under the assumptions of Theorem 3, and conditioned on the event* $\mathcal{E}^C$, *the following holds:*

$$\mathcal{T}_2 = \Sigma_{t=1}^T g_t^\nu(s_t,a_t) \leq \frac{2\beta_2 L\nu}{\iota\epsilon\tau}\Sigma_{t=1}^T\|\phi(s_t,a_t)\|_{(\Lambda_t)^{-1}}. \tag{74}$$

Now, the result of Lemma 17 enables us to utilize Lemma 11 in Abbasi-Yadkori et al. (2011) to show the sublinearity of $\mathcal{T}_2$.

**Step 3.** Now combining last two steps and upper bounding the normalized term obtained in step 2, we can apply Lemma 11 from Abbasi-Yadkori et al. (2011) to get the final result.

## K  Proof of Lemma 14

We can directly apply Theorem 2 from Abbasi-Yadkori et al. (2011) to show that each of the events $\mathcal{E}_1^C$ and $\mathcal{E}_2$ holds sepereatedly with a probability of at least $1 - \delta$. Using the union bound, we find that the event $\mathcal{E}_1^C \cap \mathcal{E}_2^C$ holds with a probability of at least $1 - 2\delta$.

## L  OPTIMISM FOR LINEAR CONTEXTUAL BANDITS

### L.1  PRELIMNINARY RESULTS

**Lemma 18** *Under Assumptions 4, 5, and 6, conditioned on the event $\mathcal{E}^C$, for all $(s,t) \in \mathcal{S} \times [T]$, we have $\alpha\phi(s, a_s^*) \in \phi(s, \mathcal{A}_t(s))$, where $\alpha \geq \epsilon$.*

The proof of this Lemma is provided in Appendix L.3.

**Lemma 19** *Let $\frac{\tau}{\tau+\iota} \leq \frac{\tau}{\tau+2\beta_2 L\|\overline{\phi(s,a_s^*)}\|_{\Lambda_t^{-1}}}$. Then, under Assumptions 4- 6, we have $\alpha\phi(s, a^*) \in \phi(s, \mathcal{A}_t(s))$, for some $\alpha \in \left[\frac{\tau}{\tau+2\beta_2 L\|\overline{\phi(s,a_s^*)}\|_{\Lambda_t^{-1}}}, 1\right]$.*

The proof of this Lemma is provided in Appendix L.3.

### L.2  PROOF OF LEMMA 15

Pick an arbitrary action $a' \in \mathcal{A}_t(s)$. Then, we will have:

$$\max_{a \in \mathcal{A}_t(s)} \langle \phi(s,a), \theta_t \rangle + b_t(s,a) \geq \langle \phi(s,a'), \theta_t \rangle + b_t(s,a')$$
$$= \langle \phi(s,a'), \theta_t \rangle + \beta_1 \|\phi(s,a')\|_{\Lambda_t^{-1}} + g_t^\nu(s,a') \tag{75}$$

Now, on the event $\mathcal{E}_2^C$ we will have:

$$\langle \phi(s,a'), \theta_t \rangle + \beta_1 \|\phi(s,a')\|_{\Lambda_t^{-1}} + g_t^\nu(s,a') \geq \langle \phi(s,a'), \theta^* \rangle + g_t^\nu(s,a'). \tag{76}$$

Thus, combining Equations 75 and 76 yields the following:

$$\max_{a \in \mathcal{A}_t(s)} \langle \phi(s,a), \theta_t \rangle + b_t(s,a) \geq \langle \phi(s,a'), \theta^* \rangle + g_t^\nu(s,a') \tag{77}$$

This brings us to analyze two sub-cases:

*Sub-case one:* Assume that $\frac{\tau}{\tau+2\beta_2 L\|\overline{\phi(s,a_s^*)}\|_{\Lambda_t^{-1}}} \leq \frac{\tau}{\tau+\iota}$. Then, by Lemma 18, there exists an action $a_\alpha \in \mathcal{A}_t(s)$ such that $\phi(s, a_\alpha) = \alpha\phi(s, a_s^*)$, where $\alpha \geq \epsilon$. Thus, by replacing $a'$ with $a_\alpha$ in Eq.(77) we have:

$$\max_{a \in \mathcal{A}_t(s)} \langle \phi(s,a), \theta_t \rangle + b_t(s,a) \geq \langle \phi(s,a_\alpha), \theta^* \rangle + g_t^\nu(s,a_\alpha) = \alpha\langle \phi(s,a^*), \theta^* \rangle + g_t^\nu(s,a_\alpha). \tag{78}$$

Now, using the definition of $g_t^\nu(.)$ in Eq.(71), and considering the fact that $\|\overline{\phi(s,a_s^*)}\|_{\Lambda_t^{-1}} = \|\overline{\alpha\phi(s,a_s^*)}\|_{\Lambda_t^{-1}}$, we have the following:

$$g_t^\nu(s,a_\alpha) = \nu \times \left(1 - \frac{\tau}{\tau + 2\beta_2 L\|\overline{\phi(s,a^*)}\|_{(\Lambda_t)^{-1}}}\right) \tag{79}$$

By setting $\nu = \frac{\tau+\iota}{\iota}$, and considering that $\frac{\tau}{\tau+2\beta_2 L\|\overline{\phi(s,a_s^*)}\|_{\Lambda_t^{-1}}} \leq \frac{\tau}{\tau+\iota}$ we obtain the following:

$$g_t^\nu(s,a_\alpha) \geq \frac{(\tau+\iota)}{\iota}\left(\frac{\iota}{\tau+\iota}\right) = 1 \geq (1-\epsilon) \tag{80}$$

Since we assumed in our problem formulation that $r(s,a) \in [0,1]$ for all $a \in \mathcal{A}$, we obtain:

$$g_t^\nu(s, a_\alpha) \geq (1 - \epsilon) \geq (1 - \epsilon)\langle\phi(s, a_s^*), \theta^*\rangle \tag{81}$$

Now, combining Equations (78) and (81) yields:

$$\max_{a \in \mathcal{A}_t(s)} \langle\phi(s, a), \theta_t\rangle + b_t(s, a) \geq \alpha\langle\phi(s, a_s^*), \theta^*\rangle + (1 - \epsilon)\langle\phi(s, a_s^*), \theta^*\rangle \geq \langle\phi(s, a_s^*), \theta^*\rangle, \tag{82}$$

where the last inequlity obtained by the fact that $\alpha \geq \epsilon$. This completes the proof for sub-case one.

*Sub-case two:* Now, assume that $\frac{\tau}{\tau + 2\beta_2 L\|\phi(s, a_s^*)\|_{\Lambda_t^{-1}}} \geq \frac{\tau}{\tau + \iota}$. Then, by Lemma 19, there exists an action $a_\alpha \in \mathcal{A}_t(s)$ such that $\phi(s, a_\alpha) = \alpha\phi(s, a_s^*)$, where $\alpha \geq \frac{\tau}{\tau + 2\beta_2 L\|\phi(s, a_s^*)\|_{(\Lambda_t)^{-1}}}$. Thus, by replacing $a'$ with $a_\alpha$ in Eq.(77), we have:

$$\max_{a \in \mathcal{A}_t(s)} \langle\phi(s, a), \theta_t\rangle + b_t(s, a) \geq \alpha\langle\phi(s, a_s^*), \theta^*\rangle + g_t^\nu(s, a_\alpha). \tag{83}$$

Now, similar to the sub-case one, by setting $\nu = \frac{\tau + \iota}{\iota} \geq 1$, we have:

$$\begin{aligned}
g_t^\nu(s, a_\alpha) &= \frac{\tau + \iota}{\iota} \times \left(1 - \frac{\tau}{\tau + 2\beta_2 L\|\overline{\phi(s, a_s^*)}\|_{(\Lambda_t)^{-1}}}\right) \\
&\geq \left(1 - \frac{\tau}{\tau + 2\beta_2 L\|\overline{\phi(s, a_s^*)}\|_{(\Lambda_t)^{-1}}}\right) \geq 1 - \alpha \geq (1 - \alpha)\langle\phi(s, a_s^*), \theta^*\rangle
\end{aligned} \tag{84}$$

where the last inequality is obtained by the fact that $r(s, a) \in [0, 1]$ for all $a \in \mathcal{A}$. Thus, combining Equations (83) and (84) yields:

$$\max_{a \in \mathcal{A}_t(s)} \langle\phi(s, a), \theta_t\rangle + b_t(s, a) \geq \alpha\langle\phi(s, a_s^*), \theta^*\rangle + (1 - \alpha)\langle\phi(s, a_s^*), \theta^*\rangle = \langle\phi(s, a_s^*), \theta^*\rangle. \tag{85}$$

This completes the proof for sub-case two as well as the Lemma 15.

### L.3 PROOF OF LEMMAS 18 AND 19

**Proof of Lemma 18:** By Assumption 6, there exists a positive number $\mu \in [\epsilon, \frac{\tau}{\sqrt{d}}]$ such that $\mu\frac{\phi(s, a_s^*)}{\|\phi(s, a_s^*)\|} \in \mathcal{F}_s$. Now, choose $\alpha = \frac{\mu}{\|\phi(s, a_s^*)\|}$. Then, we have the following:

$$\alpha\phi(s, a_s^*) = (\alpha\|\phi(s, a_s^*)\|) \times \frac{\phi(s, a_s^*)}{\|\phi(s, a_s^*)\|} = \mu\frac{\phi(s, a_s^*)}{\|\phi(s, a_s^*)\|},$$

which implies that $\alpha\phi(s, a_s^*) \in \mathcal{F}_s$. By Assumption 1, we have: $\|\phi(s, a_s^*)\| \leq L \leq 1$, which implies: $\alpha \geq \mu \geq \epsilon$. This implies that $\alpha\phi(s, a_s^*) \in \mathcal{F}_s$ for some $\alpha \geq \epsilon$. It remains to show that $\alpha\phi(s, a_s^*) \in \phi(s, \mathcal{A}_t(s))$ for all $(s, t) \in \mathcal{S} \times [T]$. Note that we can write the following:

$$\|\alpha\phi(s, a_s^*)\| = \|\mu\frac{\phi(s, a_s^*)}{\|\phi(s, a_s^*)\|}\| = \mu \leq \frac{\tau}{\sqrt{d}}, \tag{86}$$

which by Eq.(69) implies that $\alpha\phi(s, a_s^*) \in \phi(s, \mathcal{A}^{\frac{\tau}{\sqrt{d}}}(s))$. Since $\mathcal{A}^{\frac{\tau}{\sqrt{d}}}(s) \subset \mathcal{A}_t(s)$, we have: $\alpha\phi(s, a_s^*) \in \phi(s, \mathcal{A}_t(s))$ as well, i.e., there exists an $a \in \mathcal{A}_t(s)$ such that $\phi(s, a) = \alpha\phi(s, a_s^*)$, where $\alpha \geq \epsilon$. $\square$

**Proof of Lemma 19:** We decompose the proof of this lemma into two cases:

**Case 1:** Assume that $a_s^*$ does not lie on the constraint's boundary. Then, by Assumption 6 we have: $\langle \phi(s, a_s^*), \gamma^* \rangle \leq \tau - \iota$. Therefore, we can show that in this case, $a_s^* \in \mathcal{A}_t(s)$. To prove our claim, note that, on the event $\mathcal{E}^C$, we have:

$$\langle \phi(s, a_s^*), \gamma_t \rangle + \beta_2 \| \phi(s, a_s^*) \|_{\Lambda_t^{-1}} \leq \langle \phi(s, a_s^*), \gamma^* \rangle + 2\beta_2 \| \phi(s, a_s^*) \|_{\Lambda_t^{-1}}$$
$$= \tau - \iota + 2\beta_2 \| \phi(s, a_s^*) \|_{\Lambda_t^{-1}}. \tag{87}$$

On the other hand, the condition $\frac{\tau}{\tau + \iota} \leq \frac{\tau}{\tau + 2\beta_2 L \| \overline{\phi(s, a_s^*)} \|_{\Lambda_t^{-1}}}$ implies that $2\beta_2 L \| \overline{\phi(s, a_s^*)} \|_{\Lambda_t^{-1}} \leq \iota$. Thus, by Eq.(87), we have:

$$\langle \phi(s, a_s^*), \gamma_t \rangle + \beta_2 \| \phi(s, a_s^*) \|_{\Lambda_t^{-1}} \leq \tau - \iota + 2\beta_2 \| \phi(s, a_s^*) \|_{\Lambda_t^{-1}} \leq \tau$$

which implies $a_s^* \in \mathcal{A}_t^{\text{RLS}}(s) \subset \mathcal{A}_t(s)$. Thus, for $\alpha = 1$, we have $\alpha \phi(s, a_s^*) \in \phi(s, \mathcal{A}_t(s))$, which completes the proof for Case 1.

**Case 2:** Now, assume that $a_s^*$ lies on the constraint's boundry, i.e., we have $\langle \phi(s, a_s^*), \gamma^* \rangle = \tau$. We will show that $\alpha \phi(s, a_s^*) \in \phi(s, \mathcal{A}_t(s))$ for $\alpha = \frac{\tau}{\tau + 2\beta_2 L \| \overline{\phi(s, a_s^*)} \|_{\Lambda_t^{-1}}}$. To prove this, note that since $\alpha \geq \frac{\tau}{\tau + \iota}$, by Assumption 3, and because $a_s^*$ resides on the constraint's boundary, we must have $\alpha \phi(s, a_s^*) \in \phi(s, \mathcal{A})$, i.e., there exists an action $a_\alpha \in \mathcal{A}$ such that $\phi(s, a_\alpha) = \alpha \phi(s, a_s^*)$. Now, it remains to show that $a_\alpha \in \mathcal{A}_t(s)$. Conditioned on the event $\mathcal{E}^C$, we have:

$$\langle \phi(s, a_\alpha), \gamma_t \rangle + \beta_2 \| \phi(s, a_\alpha) \|_{\Lambda_t^{-1}} = \alpha \phi(s, a_s^*), \gamma_t \rangle + \beta_2 \| \phi(s, a_s^*) \|_{\Lambda_t^{-1}}$$
$$\leq \alpha \left( \langle \phi(s, a_s^*), \gamma^* \rangle + 2\beta_2 \| \phi(s, a_s^*) \|_{\Lambda_t^{-1}} \right) \leq \alpha \left( \langle \phi(s, a_s^*), \gamma^* \rangle + 2\beta_2 L \| \overline{\phi(s, a_s^*)} \|_{\Lambda_t^{-1}} \right) \tag{88}$$

Now, by substituting $\langle \phi(s, a_s^*), \gamma^* \rangle = \tau$ and $\alpha = \frac{\tau}{\tau + 2\beta_2 L \| \overline{\phi(s, a_s^*)} \|_{\Lambda_t^{-1}}}$ in Eq. (88), we have:

$$\langle \phi(s, a_\alpha), \gamma_t \rangle + \beta_2 \| \phi(s, a_\alpha) \|_{\Lambda_t^{-1}} \leq \frac{\tau}{\tau + 2\beta_2 L \| \overline{\phi(s, a_s^*)} \|_{\Lambda_t^{-1}}} \left( \tau + 2\beta_2 L \| \overline{\phi(s, a_s^*)} \|_{\Lambda_t^{-1}} \right) \leq \tau, \tag{89}$$

which implies $a_\alpha \in \mathcal{A}_t^{\text{RLS}}(s) \subset \mathcal{A}_t(s)$. This completes the proof of Case 2 and Lemma 19. $\square$

# M  PROOF OF LEMMA 16

**Proof of the Lemma 16:** By utilizing Lemma 15, we can infer:

$$\text{Regret}(T) = \Sigma_{t=1}^T \langle \phi(s_t, a_{s_t^*}), \theta^* \rangle - \langle \phi(s_t, a_t), \theta^* \rangle \leq \Sigma_{t=1}^T \langle \phi(s_t, a_t), \theta_t \rangle + b_t(s_t, a_t) - \langle \phi(s_t, a_t), \theta^* \rangle$$
$$= \Sigma_{t=1}^T \langle \phi(s_t, a_t), \theta_t - \theta^* \rangle + \beta_1 \| \phi(s_t, a_t) \|_{\Lambda_t^{-1}} + g_t^\nu(s_t, a_t) \tag{90}$$

Now, conditioned on the event $\mathcal{E}^C$, we have $\langle \phi(s_t, a_t), \theta_t - \theta^* \rangle \leq \beta_1 \| \phi(s_t, a_t) \|_{\Lambda_t^{-1}}$. Thus, we can continue Eq.(90) as follows:

$$\text{Regret}(T) \leq 2\beta_1 \Sigma_{t=1}^T \| \phi(s_t, a_t) \|_{\Lambda_t^{-1}} + \Sigma_{t=1}^T g_t^\nu(s_t, a_t) \tag{91}$$

This completes the proof. $\square$

## N    PROOF OF LEMMA 17

We begin by stating a Lemma that is helpful in the proof of Lemma 17.

**Lemma 20** *Under the setup of Theorem 3 and on the event $\mathcal{E}^C$, for every selected action $a_t$ in Algorithm 2, it holds that $\epsilon \leq \|\phi(s_t, a_t)\|, \forall t \in [T]$.*

The proof of this Lemma is provided in Appendix N.1. Now, we are ready for the proof of Lemma 17:

**Proof of Lemma 17:**    Using the expression for the $g_t^\nu(.)$ function, we have:

$$
\begin{aligned}
\Sigma_{t=1}^T g_t^\nu(s_t, a_t) &= \Sigma_{t=1}^T \nu \left( 1 - \frac{\tau}{\tau + 2\beta_2 L \|\overline{\phi(s_t, a_t)}\|_{(\Lambda_t)^{-1}}} \right) \\
&= (2\beta_2 L \nu) \Sigma_{t=1}^T \frac{\|\overline{\phi(s_t, a_t)}\|_{(\Lambda_t)^{-1}}}{\tau + 2\beta_2 L \|\overline{\phi(s_t, a_t)}\|_{(\Lambda_t)^{-1}}} \leq \frac{(2\beta_2 L \nu)}{\tau} \Sigma_{t=1}^T \|\overline{\phi(s_t, a_t)}\|_{(\Lambda_t)^{-1}},
\end{aligned}
\tag{92}
$$

where the last inequality follows from the fact that $\tau \leq \tau + 2\beta_2 L \|\overline{\phi(s_t, a_t)}\|_{(\Lambda_t)^{-1}}$. Now, since $\|\overline{\phi(s_t, a_t)}\|_{(\Lambda_t)^{-1}} = \|\frac{\phi(s_t, a_t)}{\|\phi(s_t, a_t)\|}\|_{(\Lambda_t)^{-1}}$, we can apply Lemma 20 as follows:

$$
\Sigma_{t=1}^T g_t^\nu(s_t, a_t) \leq \frac{(2\beta_2 L \nu)}{\tau} \Sigma_{t=1}^T \|\overline{\phi(s_t, a_t)}\|_{(\Lambda_t)^{-1}} \leq \frac{(2\beta_2 L \nu)}{\epsilon \tau} \Sigma_{t=1}^T \|\phi(s_t, a_t)\|_{\Lambda_t^{-1}},
\tag{93}
$$

which completes the proof. $\square$

### N.1    PROOF OF LEMMA 20

To prove Lemma 20, we utilize a contradiction strategy. Assume that for some $t \in [T]$, we have $\|\phi(s_t, a_t)\| < \epsilon$. Note that by Assumption 6, we have $\alpha \frac{\phi(s_t, a_t)}{\|\phi(s_t, a_t)\|} \in \phi(s_t, \mathcal{A})$, for some $\alpha \in [\epsilon, \frac{\tau}{\sqrt{d}}]$. Also, since $\alpha \leq \frac{\tau}{\sqrt{d}}$, we have: $\alpha \frac{\phi(s_t, a_t)}{\|\phi(s_t, a_t)\|} \in \phi(s_t, \mathcal{A}^{\frac{\tau}{\sqrt{d}}}(s)) \subset \phi(s_t, \mathcal{A}_t(s))$, i.e., there exists an $a' \in \mathcal{A}_t(s)$ such that $\phi(s, a') = \alpha \frac{\phi(s_t, a_t)}{\|\phi(s, a_t)\|}$. Now, we show that at time $t$, Algorithm 2 will prefer to choose $a'$ instead of $a_t$, which results in a contradiction and completes the proof. To show that the algorithm prefers to choose $a'$, we consider the following:

$$
\langle \phi(s_t, a_t), \theta_t \rangle + b_t(s_t, a_t) = \langle \phi(s_t, a_t), \theta_t \rangle + \beta_1 \|\phi(s_t, a_t)\|_{\Lambda_t^{-1}} + g_t^\nu(s_t, a_t)
\tag{94}
$$

Now, conditioned on the event $\mathcal{E}^C$, we have $r(s_t, a_t) = \langle \phi(s_t, a_t), \theta^* \rangle \leq \langle \phi(s_t, a_t), \theta_t \rangle + \beta_1 \|\phi(s_t, a_t)\|_{\Lambda_t^{-1}}$. Since in our problem formulation we assume that $r(.) \in [0, 1]$, we can say the following:

$$
0 \leq r(s_t, a_t) \leq \langle \phi(s_t, a_t), \theta_t \rangle + \beta_1 \|\phi(s_t, a_t)\|_{\Lambda_t^{-1}} \leq \frac{\epsilon}{\|\phi(s_t, a_t)\|} \left( \langle \phi(s_t, a_t), \theta_t \rangle + \beta_1 \|\phi(s_t, a_t)\|_{\Lambda_t^{-1}} \right),
\tag{95}
$$

where the last inequality follows from the fact that, according to the contradiction assumption, we have $\|\phi(s_t, a_t)\| \leq \epsilon$, which implies $1 \leq \frac{\epsilon}{\|\phi(s_t, a_t)\|}$. Now, combining Equations (94) and (95), we have:

$$
\langle \phi(s_t, a_t), \theta_t \rangle + b_t(s_t, a_t) \leq \frac{\alpha}{\|\phi(s_t, a_t)\|} \left( \langle \phi(s_t, a_t), \theta_t \rangle + \beta_1 \|\phi(s_t, a_t)\|_{\Lambda_t^{-1}} \right) + g_t^\nu(s_t, a_t)
\tag{96}
$$

Now, substituting the definition of $\phi(s_t, a') = \alpha \frac{\phi(s_t, a_t)}{\|\phi(s_t, a_t)\|}$ into Eq. (96), we get:

$$\langle\phi(s_t, a_t), \theta_t\rangle + b_t(s_t, a_t) \le \langle\phi(s_t, a'), \theta_t\rangle + \beta_1\|\phi(s_t, a')\|_{\Lambda_t^{-1}} + g_t^\nu(s_t, a_t) \tag{97}$$

Now, using the definition of $g_t^\nu$ in Eq.(71), one can verify that $g_t^\nu(s_t, a') = g_t^\nu(s_t, a_t)$. Thus, we can continue Eq. 97 as follows:

$$\langle\phi(s_t, a_t), \theta_t\rangle + b_t(s_t, a_t) \le \langle\phi(s_t, a'), \theta_t\rangle + b_t(s_t, a'). \tag{98}$$

The last inequality in Eq. (98) implies that in line 6 of Algorithm 2, action $a'$ is preferred to action $a_t$, which is contradiction and proves that $\|\phi(s_t, a_t)\| \ge \epsilon$, for all $t \in [T]$. $\square$

## O    PROOF OF THEOREM 3

We apply Lemmas 16 and 17 to get the following:

$$\text{Regret}(T) \le \Sigma_{t=1}^T (2\beta_1 + \frac{2\beta_2 L\nu}{\iota\epsilon\tau})\|\phi(s_t, a_t)\|_{\Lambda_t^{-1}} \tag{99}$$

Following the steps outlined in the proof of Theorem 3 in Abbasi-Yadkori et al. (2011), we proceed as follows:

$$\Sigma_{t=1}^T \|\phi(s_t, a_t)\|_{\Lambda_t^{-1}} \le \sqrt{T\Sigma_{t=1}^T \|\phi(s_t, a_t)\|_{\Lambda_t^{-1}}^2} \tag{100}$$

By Assumption 4, given that $L \le 1$ and $\lambda = 1$ in Algorithm 2, it can be shown that $\|\phi(s_t, a_t)\|_{\Lambda_t^{-1}} \le 1$. Consequently, the following inequality holds:

$$\|\phi(s_t, a_t)\|_{\Lambda_t^{-1}}^2 \le 2\log(1 + \|\phi(s_t, a_t)\|_{\Lambda_t^{-1}}^2). \tag{101}$$

Thus, by Equations 100 and 101 we have:

$$\sqrt{T\Sigma_{t=1}^T \|\phi(s_t, a_t)\|_{\Lambda_t^{-1}}^2} \le \sqrt{2T\Sigma_{t=1}^T \log(1 + \|\phi(s_t, a_t)\|_{\Lambda_t^{-1}}^2)}$$
$$= \sqrt{2T(\log(\det(\Lambda_T)) - \log(\lambda^d))}. \tag{102}$$

where the last inequality is obtained by Lemma 11 from Abbasi-Yadkori et al. (2011). Now, considering that $\|\phi(s_t, a_t)\| \le L$, it follows that the trace of $\Lambda_T$ is bounded by $d\lambda + TL^2$. Since $\Lambda_T$ is a positive definite matrix, the determinant of $\Lambda_T$ can be bounded by:

$$det(\Lambda_T) \le (\frac{\text{trace}(\Lambda_T)}{d})^d \le (\frac{d\lambda + TL^2}{d})^d.$$

Combining everything together yields:

$$\sqrt{2T(\log(\det(\Lambda_T)) - \log(\lambda^d))} \le \sqrt{2T(\log((\frac{d\lambda + TL^2}{d})^d) - \log(\lambda^d))}$$
$$= \sqrt{2Td\log((\frac{d\lambda + TL^2}{\lambda d}))} = \sqrt{2Td\log((\frac{d\lambda + TL^2}{\lambda d}))} \tag{103}$$

Now, utilizing Equations (100) through (103), we conclude that:

$$\Sigma_{t=1}^T \|\phi(s_t, a_t)\|_{\Lambda_t^{-1}} \le \sqrt{2Td\log((\frac{d\lambda + TL^2}{\lambda d}))} \tag{104}$$

Now, by integrating Equations (99) and (104), we establish the desired upper bound as follows:

$$\text{Regret}(T) \le (2\beta_1 + \frac{2\beta_2 L\nu}{\epsilon\iota\tau}) \sqrt{2Td\log((\frac{d\lambda + TL^2}{\lambda d}))}\square \tag{105}$$

# P OTHER SIMULATION SCENARIOS

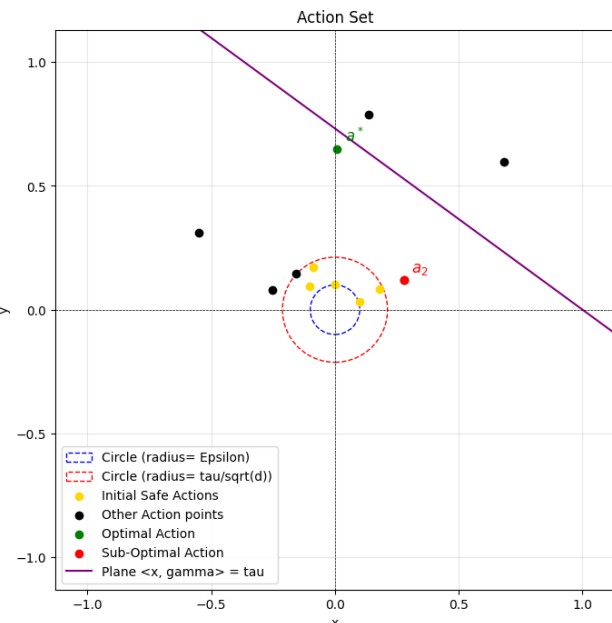

Figure 6: Action Space $\mathcal{A}$, where yellow points represent the initial safe actions, the green point $a^*$ denotes the optimal action, and the red point $a_2$ indicates the sub-optimal action. The plane $\langle \gamma^*, x \rangle = \tau$ represents the constraint boundary.

We conducted an additional experiment to evaluate the performance of NCS-LUCB compared to LC-LUCB from Pacchiano et al. (2024). The setup includes $\theta^* = [1.0, 1.0]^\top$, $\gamma^* = [0.3, 0.41]^\top$, $\tau = 0.3$, and the following action set (also depicted in Figure 6):

$$
\mathcal{A} = \left\{ \begin{bmatrix} 0.1817 \\ 0.0816 \end{bmatrix}, \begin{bmatrix} -0.1014 \\ 0.0930 \end{bmatrix}, \begin{bmatrix} 0.0014 \\ 0.1000 \end{bmatrix}, \begin{bmatrix} -0.0889 \\ 0.1723 \end{bmatrix}, \begin{bmatrix} 0.1003 \\ 0.0305 \end{bmatrix}, \begin{bmatrix} -0.5500 \\ 0.3120 \end{bmatrix}, \begin{bmatrix} -0.1586 \\ 0.1468 \end{bmatrix}, \right.
$$
$$
\left. \begin{bmatrix} 0.1362 \\ 0.7862 \end{bmatrix}, \begin{bmatrix} 0.6816 \\ 0.5962 \end{bmatrix}, \begin{bmatrix} -0.2521 \\ 0.0807 \end{bmatrix}, \begin{bmatrix} 0.2800 \\ 0.1200 \end{bmatrix}, \begin{bmatrix} 0.0093 \\ 0.6499 \end{bmatrix} \right\}.
$$

$$(106)$$

This implies that the optimal safe action is $a^* = \begin{bmatrix} 0.0093 \\ 0.6499 \end{bmatrix}$. Our algorithm, NCS-LUCB, effectively expands the estimated safe set toward the optimal action $a^*$ (see Figure 7a), achieving sublinear regret (see Figure 8a). In contrast, LC-LUCB fails to include $a^*$ in its estimated safe set. Instead, it predominantly samples from the suboptimal action $a_2 = \begin{bmatrix} 0.28 \\ 0.12 \end{bmatrix}$, resulting in a biased safe set expansion (see Figure 7b) and linear regret (see Figure 8b). All unspecified parameters are consistent with those used in the previous experiment.

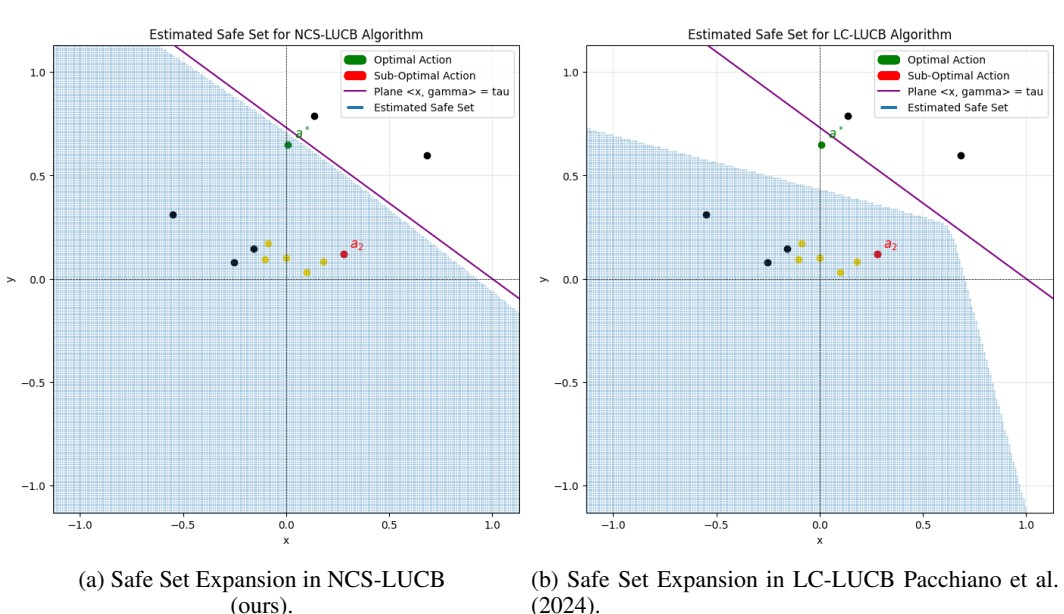

(a) Safe Set Expansion in NCS-LUCB (ours).

(b) Safe Set Expansion in LC-LUCB Pacchiano et al. (2024).

Figure 7: Blue-highlighted regions depict the estimated safe set expansion in both settings. NCS-LUCB (ours) successfully expands the estimated safe set toward the optimal point $a^*$, while LC-LUCB from Pacchiano et al. (2024) is biased toward the suboptimal action $a_2$ and fails to include the optimal point $a^*$ in the safe set.

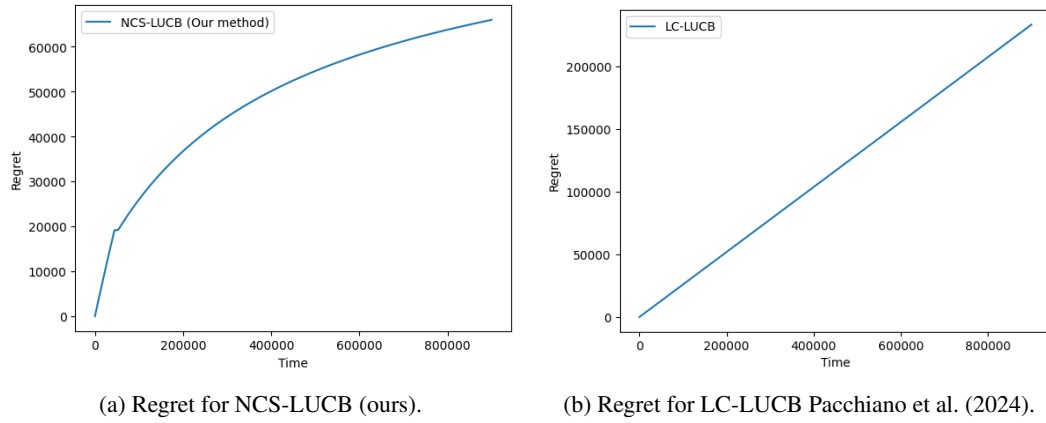

(a) Regret for NCS-LUCB (ours).

(b) Regret for LC-LUCB Pacchiano et al. (2024).

Figure 8: Comparison of the regret for NCS-LUCB (our method) and LC-LUCB inPacchiano et al. (2024).

