# OpenReview forum: "Provably Efficient Linear Bandits with Instantaneous Constraints in Non-Convex Feature Spaces"
_ICLR.cc/2025/Conference — Submitted to ICLR 2025_

### Official Review · Reviewer_8CHn · 2024-10-29

**Soundness:** 3
**Presentation:** 3
**Contribution:** 2
**Rating:** 3
**Confidence:** 3

**Summary:**

This paper extends the stage-wise safe linear bandits to the non-convex spaces. Compared with the prior work Pacchiano et al. (2024), which studies the stage-wise safe linear bandits under the assumption that the arm set is convex, this paper finds the limitation of the algorithm LC-LUCB when the arm set is a non-convex spaces. By redesigning the bonus term in the confidence radius, the improved algorithm NCS-LUCB is capable of dealing with non-convex arm set.

**Strengths:**

- The paper is clear and is easy to follow.
- The problem of the bonus term in the previous literature in the context of non-convex arm set is well explained via examples and the intuition for the new bonus term is also clear.
- Both problem-independent upper bound and lower bound are provided, showing the importance of the parameters $\epsilon,\iota$ in the problem.

**Weaknesses:**

**Majors**:
- In Line 376, the BOB method is only **adopted** by Zhao et al. (2020), and it is firstly **proposed** by [1].
- The technical contribution is limited. While the introduction of the new bonus term can be important, the rest of the analysis is quite standard.
- The proposed bonus term consists of $\iota$, which appears in Assumption 3 and is not known in practice. While it mentions the BOB technique [1] may be adopted to solve, it does not further investigate on it. As the design of the bonus term is the main technical contribution of this paper and the parameters $\iota$ plays an important role in the bounds, it is expected that the authors can provide a complete solution towards the bonus design.

**Minors**:
- It would be great to show some more complex examples (at least in the experiment). As this paper deals with linear bandits, the proposed examples, including the toy examples and the experimental examples, are composed by arms along the axis, making them quite similar to K-armed bandits (with minor additional setups).

[1] Wang Chi Cheung, David Simchi-Levi, Ruihao Zhu. _Proceedings of the Twenty-Second International Conference on Artificial Intelligence and Statistics_, PMLR 89:1079-1087, 2019.

**Questions:**

- In Assumption 1, it is assumed that $\max(\|\theta^*\|, \|\gamma^*\|) \leq \sqrt{d}$. Is it possible to bound it by other notations, e.g., $B$? In this case, the dependence on this quantity can be better reflected in the final regret bound.
- Is it possible to provide problem-dependent upper and lower bounds, consisting of the reward and cost gaps?
- According to Pacchiano et al. (2024), the minimax lower bound for the convex case is $\max \left( \frac{d \sqrt{T}}{8 e^2}, \frac{1 - r_0}{21 (\tau - c_0)^2} \right)$ which is of order $\Omega(d\sqrt{T})$. And Theorem 2 suggests the minimax lower bound in the nonconvex case is $\max \left\\{ \frac{1}{27} \sqrt{(d - 1) T}, \frac{1 - 2\epsilon}{\epsilon} \left( \frac{1 - \iota}{\iota} \right)^2 \right\\}$ which is of order $\Omega(\sqrt{dT})$. It is expected that the nonconvex case is a harder problem, thus, the lower bound should be larger. Can the author comment on this? In addition, how does $\tau$, the threshold on the cost, influence the lower bound?
- Is it possible to show the price (additional regret) for the nonconvexity in the arm set, compared to the convex arm set case?
- Can the authors kindly compare  this work with Amani et al. (2019), Pacchiano et al. (2024), and Moradipar et al. (2021)?

**Details Of Ethics Concerns:**

None.

---

> ### Author Response · Authors · 2024-11-21
>
> **Q.1** The technical contribution is limited.
>
> **A.1**  Thank you for the feedback. Expanding the arm space from star-convexity to a local point assumption is important for several reasons:
>
> **Why Relax Star-Convexity:** Star-convexity often does not hold in practical settings with irregular decision sets or lacking global geometric properties. Examples include discrete action spaces, the VC example, and the smart building scenario discussed in our paper (Lines 254-265,  697-712), which inherently lack star-convexity. Similarly, robotics and autonomous vehicle problems do not satisfy this condition due to constraints like collision avoidance and traffic rules (Section C.2, revised paper). Relaxing this assumption extends the applicability of our methods to a broader range of real-world problems.
>
> **Challenges of Relaxation:** Without star-convexity, efficient exploration becomes more challenging due to the absence of global structure. Please note that in non-star-convex cases there is no line connecting the safe action to the optimal action within the action space. As a result, the agent may lack motivation to explore the optimal direction (unlike in the star-convexity scenario) to expand the estimated safe set toward the optimal direction. Hence, traditional exploration strategies may not perform well in unstructured decision sets and result in linear regret (please see Figure 1 in our paper).
>
> **Our Contribution:**  Removing the star-convexity assumption introduces substantial challenges, as key results from prior work, such as the fact that the **Optimism lemma** in Pacchiano et al. (2024), no longer holds in our setting. To address this issue, our method introduces a new bonus term, $g_t^\nu$ (Equation 4), which normalizes feature vectors to prioritize **uncertainty reduction** across all directions rather than focusing solely on expected rewards.  Using our bonus, we developed new theoretical tools, including a novel **Optimism lemma specifically designed for non-star-convex** decision sets (see **Lemmas 2, 5, 6**). Additionally, we proved that the **regret resulting from our new bonus remains sublinear** (see **Lemmas 4 and 7, and Theorem 1** in our paper). These contributions represent significant theoretical advancements that go beyond the introduction of the new bonus term and are critical for tackling non-star-convex settings effectively.
>
> Our approach not only addresses these challenges but also offers insights applicable to unconstrained bandit settings, potentially overcoming some weaknesses of UCB highlighted in [1]. We believe further research into this strategy is promising.
>
>
> **Q.2** While the introduction of the new bonus term can be important, the rest of the analysis is quite standard.
>
> **A.2** We respectfully disagree that the rest of our analysis is standard. Removing the star-convexity assumption presents significant challenges because key results from prior work, such as the **Optimism lemma in Pacchiano et al. (2024), no longer hold in our setting**. To overcome this, we developed new theoretical tools, **including a novel optimism lemma tailored for non-star-convex decision sets (see Lemmas 2,5,6)**. Furthermore, we demonstrate that the **regret resulting from our new bonus remains sublinear** (see **Lemmas 4, and 7, and Theorem 1** in our paper). These contributions involve significant theoretical advancements beyond the introduction of the new bonus term.
>
> **Q.3** In Assumption 1, it is assumed that $\max(|\theta^*|, |\gamma^*|) \leq \sqrt{d}$. Is it possible to bound it by other notations, e.g., $B$? In this case, the dependence on this quantity can be better reflected in the final regret bound.
>
> **A.3**  Using $B$ notation only affects the definitions of $\beta_1$ and $\beta_2$, e.g., $\beta_1 = \beta_2 = \sigma \sqrt{d \log\left(\frac{1 + \frac{T L^2}{\lambda}}{\delta}\right)} + \sqrt{\lambda} B$. However, the dependency of the upper bound on $d$ remains unchanged.
>
> **Q.4** According to Pacchiano et al. (2024), the minimax lower bound for the convex case is $\max (\frac{d\sqrt{T}}{8e^2}, \frac{1 - r_0}{21(\tau - c_0)^2})$, which is of order $\Omega(d\sqrt{T})$. And Theorem 2 suggests the minimax lower bound in the nonconvex case is $\max (\frac{1}{27}\sqrt{(d-1)T}, \frac{1 - 2\iota}{\epsilon\iota^2})$, which is of order $\Omega(\sqrt{dT})$. It is expected that the nonconvex case is a harder problem; thus, the lower bound should be larger. Can the author comment on this?
>
>
>
> **A.4**  We apologize for the confusion. There is an error in how we stated the lower bound, which, as the reviewer correctly noted, should be $\max( \frac{d}{8 e^2}\sqrt{T}, \, \frac{1-2 \epsilon}{\epsilon} (\frac{1-\iota}{\iota})^2)$ (please see lines 31, 128, 397-400 in our revised paper).
>
> **Q.5** How does $\tau$, influence the lower bound?
>
> **A.5** According to Shi et al. (2023), the lower bound is inversely proportional to the square of the safety gap $\frac{1}{\tau^2}$.

---

> ### Author Response · Authors · 2024-11-21
>
> **Q.6**  Is it possible to show the price (additional regret) for the nonconvexity in the arm set, compared to the convex arm set case?
>
> **A.6**  Yes, our regret upper bound is $\tilde{\mathcal{O}}\big( d (1 + \frac{\tau}{\epsilon \iota}) \sqrt{T} \big)$, which includes an additional $\frac{1}{\epsilon \iota}$ factor compared to convex and star-convex cases. Thus, the non-convexity in our problem contributes the $\frac{1}{\epsilon \iota}$ term to the upper bound.
>
> **Q.7** Can the authors kindly compare this work with Amani et al. (2019), Pacchiano et al. (2024), and Moradipar et al. (2021)?
>
> **A.7** Thank you for your question. In lines 665–682 of our original paper (see lines 136-150 of our revised paper), we provided a detailed discussion. Here is a summary:
> Amani et al. (2019) studied linear bandits with a convex decision set and proposed a UCB-based method with two phases: pure safe search and safe exploration-exploitation, achieving a regret of $\tilde{O}(T^{\frac{2}{3}})$. Pacchiano et al. (2024) addressed the problem under the star-convexity assumption, removing the pure-exploration phase and achieving a regret of $\tilde{O}(d \frac{\sqrt{T}}{\tau})$, along with a lower bound showing dependency on the safety gap. Moradipari et al. (2021) also assumed a star-convex decision set and applied Thompson Sampling, achieving a regret of $\tilde{O}(d^{\frac{3}{2}} \frac{\sqrt{T}}{\tau})$.
>
>
> **Q.8** Is it possible to provide problem-dependent upper and lower bounds, consisting of the reward and cost gaps?
>
>
> **A.8** The cost gaps of the initial safe action $\tau$ and the optimal action $\iota$ are included in our upper bound.
> Providing problem-dependent bounds that include the reward gap is an interesting direction for future work. Our novel bonus term $g_t^\nu$ normalizes feature vectors to prioritize uncertainty reduction across all directions, rather than focusing solely on expected rewards. This approach could potentially make the problem more robust with respect to the reward gap. However, further study is required to analyze the exact effect of our method in this context.
>
> **Q.9** The proposed bonus term consists of $\iota$, which appears in Assumption 3 and is not known in practice. While it mentions the BOB technique [1] may be adopted to solve, it does not further investigate on it. As the design of the bonus term is the main technical contribution of this paper and the parameter $\iota$ plays an important role in the bounds, it is expected that the authors can provide a complete solution towards the bonus design.
>
>
> **A.9** In practice, $\nu$ can be treated as a tunable hyperparameter, adjusted using domain knowledge or standard hyperparameter tuning methods commonly used in optimization and machine learning (see Amani et al. (2019) for a discussion on the unknown safety gap). Our main contribution demonstrates that setting $\nu \geq \frac{\tau}{\tau + \iota}$ ensures sublinear regret while maintaining safety guarantees. Importantly, the exact value of $\iota$ is not needed; any $\nu$ above this threshold is sufficient.
>
>
>
> In non-star-convex settings, designing a bonus term to achieve sublinear regret is challenging, even when \(\iota\) is known. The BOB technique [1] is proposed to handle cases where $\iota$ is unknown. It does not modify the bonus term to achieve sublinear regret directly. Instead, once we have determined a bonus term that ensures sublinear regret and safety guarantees for a given $\iota$, the BOB technique can be employed to adapt the bonus term for cases where $\iota$ is unknown, thereby providing a practical solution for setting the hyperparameter.
>
>
> **Q.10** In Line 376, the BOB method is only adopted by Zhao et al. (2020), and it is firstly proposed by [1].
>
> **A.10** We appreciate the reviewer’s observation. You are correct that the BOB method was first proposed by [1], and its adoption by Zhao et al. (2020) builds upon that work. We apologize for any lack of clarity in our phrasing on Line 376 of our original submission. In the revised version, we have clarified this point (please see Line 389 in the updated paper).
>
> **Q.11** It would be great to show some more complex examples (at least in the experiment). As this paper deals with linear bandits, the proposed examples, are composed by arms along the axis, making them quite similar to K-armed bandits.
>
> **A.11**  Thank you for the suggestion. One of the main contributions of our work is addressing safe linear bandits with discrete action spaces, so it is valid that our setup resembles $K$-armed bandits. This design allows us to clearly demonstrate the theoretical properties of our method and validate the results. However, our approach is applicable to any scenario that satisfies Assumption 3, not limited to axis-aligned arms.
>
> In our final submission, we will include a more complex simulation scenario to further illustrate the robustness and generality of our method.

---

> ### Author Response · Authors · 2024-11-21
>
> Finally, we thank the reviewer for their helpful comments. If our response resolves your concerns, we kindly ask you to consider raising the rating of our work. We are happy to address any further questions during the discussion period
>
> [2] Russo, D., & Van Roy, B. (2014). Learning to optimize via information-directed sampling. *Advances in Neural Information Processing Systems*, 27.

---

> ### Author Response · Authors · 2024-11-25
>
> Dear Reviewer,
>
> As the author-reviewer discussion period is nearing its end, we would appreciate it if you could review our responses to your comments. This way, if you have further questions and comments, we can still reply before the author-reviewer discussion period ends. If our response resolves your concerns, we kindly ask you to consider raising the rating of our work. Thank you very much for your time and effort.

---

> > ### Comment · Reviewer_8CHn · 2024-11-26
> >
> > Thank the authors for the detailed reply! Most of concerns have been properly resolved. There are some left
> >
> > - Regarding the additional regret Q6, the additional regret in the upper bound is much larger than that in the lower bound result (which takes the maximum of two quantities, instead of a direct factor). Does this indicate the upper bound can be quite loose? In other words, while the authors claim that the important roles of $\epsilon$ and $\iota$ in line129, they can be dominated by $O(d\sqrt{T})$ in the lower bound, but they always appear in the upper bound result.
> > - Regarding the knowledge of instance-dependent parameters Q9: The knowledge about the instance parameters can be a weakness. While the authors state that setting $\nu \geq \frac{\tau}{\tau + \iota}$ can guarantee sublinear regret and maintain the safety guarantees, this threshold is not revealed in practice.
> >
> > Intuitively, if one arm is not identified as safe but has high reward, the algorithm should allocate some arm pulls to explore its safety. In the convex set case, this is guaranteed by the bonus term. The original bonus term in the convex case is not applicable in the nonconvex case. The main contribution of this paper is the redesign of the bonus term, which can adapt the previous algorithms to the nonconvex arm set scenario.
> >
> > However, the proposed algorithm requires instance-dependent parameters $\iota$, which is of great importance in the algorithm design and analysis. Additionally, while the paper shows the essential role of $\epsilon$ and $\iota$ in the problem, there is a large gap between the minimax upper bound and lower bound in these two parameters.
> >
> > Based on the feedback and the above evaluations, I retain my previous score.

---

> > > ### Author Response · Authors · 2024-11-27
> > >
> > > **Q.12**  Regarding the knowledge of instance-dependent parameters Q9: The knowledge about the instance parameters can be a weakness. While the authors state that setting $\nu \geq \frac{\tau}{\tau + \iota}$ can guarantee sublinear regret and maintain the safety guarantees, this threshold is not revealed in practice.
> > > Intuitively, if one arm is not identified as safe but has high reward, the algorithm should allocate some arm pulls to explore its safety. In the convex set case, this is guaranteed by the bonus term. The original bonus term in the convex case is not applicable in the nonconvex case. The main contribution of this paper is the redesign of the bonus term, which can adapt the previous algorithms to the nonconvex arm set scenario.
> > > However, the proposed algorithm requires instance-dependent parameters $\iota$, which is of great importance in the algorithm design and analysis.
> > >
> > > **A.12** We acknowledge that $\iota$ is an instance-dependent parameter that may not be known in practice. To implement our algorithm, standard hyperparameter tuning methods—such as Bayesian methods or Bandits over Bandits (as discussed in lines 387-392 of our paper)—can be employed to adjust $\iota$ accordingly. While further analysis of hyperparameter tuning for our method remains an open problem, we plan to investigate this in future work.
> > >
> > >
> > > Hyperparameter tuning is a common practice in the literature. For instance, Amani et al. (2019) rely on hyperparameter tuning for their implementation, while their analysis assumes the availability of properly chosen hyperparameters. Similarly, in other fields like optimization, convergence results often depend on carefully designed learning rates. In practice, setting the exact learning rate is not always possible, as it often depends on knowing parameters like the function's Lipschitz constant, which may be unavailable. Instead, hyperparameter tuning is commonly used to approximate effective learning rates.
> > >
> > >
> > > **Q.13** Regarding the additional regret Q6, the additional regret in the upper bound is much larger than that in the lower bound result (which takes the maximum of two quantities, instead of a direct factor). Does this indicate the upper bound can be quite loose?
> > >
> > >
> > > **A.13** While our lower bound does not directly establish the tightness of the upper bound, it includes only a constant factor in front of $\sqrt{T}$.
> > >
> > > The main purpose of our lower bound is to emphasize the critical role of $\epsilon$ and $\iota$ in the algorithm's performance. It shows that these parameters must remain non-zero; otherwise, as $\epsilon$ or $\iota$ approaches zero, the lower bound diverges to infinity, leading to linear regret. This demonstrates the necessity of ensuring non-zero values for $\epsilon$ and $\iota$ to achieve sublinear regret.
> > >
> > > Moreover, the structure of our lower bound—taking the maximum of two terms (e.g., $ \max(d \sqrt{T}, \frac{1}{\epsilon \iota^2})$)—is common in the literature. It reflects the distinct contributions of factors like dimensionality $d$ and $\epsilon$ and $\iota$ to the regret bound. Similar dependencies on safety parameters can be observed in works such as Pacchiano et al. (2024), Shi et al. (2023), and Pacchiano et al. (2021).
> > >
> > > **Q.14** In other words, while the authors claim that the important roles of $\epsilon$ and $\iota$ in line129, they can be dominated by $\mathcal{O}(d\sqrt{T})$ in the lower bound, but they always appear in the upper bound result.
> > >
> > > **A.14**  When $\epsilon$ and $\iota$ are small, the term $\frac{1}{\epsilon \iota^2}$ in our lower bound becomes significant, dominating the $\mathcal{O}(d\sqrt{T})$ term unless $T$ is very large. This behavior is consistent with our upper bound result.

---

> ### Author Response · Authors · 2024-11-27
> **Additional Experiment**
>
> To address the reviewer's concern about the simplicity of our experiments(please see Q.11), we have added a new section in the Appendix (Appendix P), presenting a more complex experimental setup (see Figure 6). In this experiment, we explore scenarios beyond those with arms aligned along the axes. The results demonstrate that our proposed method, NCS-LUCB, successfully expands the estimated safe set to include the optimal safe action while achieving sublinear regret. In contrast, the method by Pacchiano et al. (2024), LC-LUCB, is biased toward a suboptimal action and fails to expand the estimated safe set to include the optimal action, resulting in linear regret (see Figures 7 and 8).

---

> ### Author Response · Authors · 2024-12-02
>
> Dear Reviewer,
> As the author-reviewer discussion period is nearing its end, we would appreciate it if you could review our responses to your comments. This way, if you have further questions and comments, we can still reply before the author-reviewer discussion period ends. Thank you for your time and effort.

---

### Official Review · Reviewer_aNwp · 2024-10-29

**Soundness:** 3
**Presentation:** 3
**Contribution:** 2
**Rating:** 6
**Confidence:** 2

**Summary:**

This paper investigates safe linear bandits in non-convex feature spaces, expanding the arm space from star-convex to local-point. The primary innovative technique is the newly proposed bonus term in Equation (4).

**Strengths:**

1. The paper is well-written, particularly in its comprehensive discussion of key points.

2. Providing both upper and lower bounds simultaneously enhances the completeness of this work and renders the newly proposed assumptions more convincing.

**Weaknesses:**

1. Expanding the arm space from star-convex to local point assumption offers limited contributions.

2. The technological innovation primarily lies in the newly proposed bonus term (4).

**Questions:**

1. The lower bound is of order $\Omega(1/\iota^2)$, while the upper bound is of order $O(1/\iota)$. This appears to be contradictory, as $1/\iota^2 > 1/\iota$. Could you please clarify this discrepancy?

2. Lines 324-328: Please provide a more detailed explanation of why $\alpha\phi(a_*) \in \phi(A_t)$ implies that the distance between $\alpha\phi(a_*)$ and $\phi(A_t)$ is less than $g_t^1(a)$.

3. How did you find Assumption 3? Were there any related works that inspired this assumption?

Although I am familiar with bandits, I have not yet encountered safe bandits. Therefore, I will adjust my score based on discussions and feedback from other reviewers.

---

> ### Author Response · Authors · 2024-11-21
>
> We thank the reviewer for providing the constructive review.
>
> **Q.1** The lower bound is of order $\Omega(1 / \iota^2)$, while the upper bound is of order $O(1 / \iota)$. This appears to be contradictory, as $1 / \iota^2 > 1 / \iota$. Could you please clarify this discrepancy?
>
> **A.1**  Thank you. As explained in Remark 3 (Lines 403–405 in the revised paper), the difference arises because Theorem 2 (lower bound) applies when $K \geq \frac{32e}{\epsilon \iota^2}$. Since the upper bound of the regret depends on $\sqrt{K}$, it is clear that the upper bound exceeds the lower bound for $K \geq \frac{32e}{\epsilon \iota^2}$.
>
>
> **Q.2** Lines 324–328: Please provide a more detailed explanation of why $\alpha \phi(a_*) \in \phi(A_t)$ implies that the distance between $\alpha \phi(a_*)$ and $\phi(\mathcal{A}_t)$ is less than $g_t^1(a)$.
>
> **A.2** We apologize for the confusion. There were typos in lines 324–330 of our paper, which likely contributed to the reviewer’s confusion. We have corrected these errors in our revised paper (see lines 338–341), as follows:
> One can show that $\alpha \phi(a^*) \in \phi(A_t)$ holds for some $\alpha \geq 1-g_t^1(a^*)$. Consequently, the distance between $\phi(a^*)$ and $\phi(A_t)$ is less than $\left(1 - \frac{\tau}{\tau + 2 \beta_2 L \|\overline{\phi(a^*)}\|_{(\Lambda_t)^{-1}}}\right)$.
> We thank the reviewer for bringing this to our attention.
>
>
> **Q.3** How did you find Assumption 3? Were there any related works that inspired this assumption?
>
> **A.3** Thank you for your question.
>
> **How We Found Assumption 3:** We theorized that if an agent can safely explore within a local neighborhood of the initial safe point, it can gather enough information to make informed decisions about exploring further. This insight led us to formalize Assumption 3, ensuring the existence of a small local set where safe exploration and learning are possible without relying on global convexity.
>
> Our goal was to relax the restrictive star-convexity condition commonly used in safe exploration literature. While star-convexity, as used in Pacchiano et al. (2024), facilitates theoretical analysis, it is often too strong and unrealistic for many practical applications, especially in non-convex environments.
>
> **Practical Relevance:** Assumption 3 is motivated by scenarios discussed in our paper (e.g., discrete cases, smart building and the VC example in Lines 254-265,  697-712) and real-world challenges, such as autonomous vehicle control. The autonomous vehicle problem is inherently non-convex and non-star-convex due to collision avoidance constraints (Please refer to Section C.2 in the revised version of our paper that we uploaded). For instance, when driving behind another car, slightly increasing speed to overtake may extend the maneuver duration, causing the vehicle to remain in the opposing lane unsafely. However, the agent can safely explore the car's dynamics by driving at low speeds and, once confident, transition to high-speed maneuvers for effective overtaking.
>
> **Related Works That Inspired This Assumption:** Assumption 3 in our work is an important generalization of the star-convexity assumption (Moradipari et al. (2021), Pacchiano et al. (2024)) to tackle these challenges.
>
> In summary, Assumption 3 was developed through our efforts to enable safe exploration in non-convex environments by leveraging local exploration strategies without depending on global geometric properties.

---

> ### Author Response · Authors · 2024-11-21
>
> **Q.4**  Expanding the arm space from star-convex to local point assumption offers limited contributions.The technological innovation primarily lies in the newly proposed bonus term (4).
>
> **A.4**  Thank you for the feedback. Expanding the arm space from star-convexity to a local point assumption is important for several reasons:
>
> **Why Relax Star-Convexity:** Star-convexity often does not hold in practical settings with irregular decision sets or lacking global geometric properties. Examples include discrete action spaces, the VC example, and the smart building scenario discussed in our paper (Lines 254-265,  697-712), which inherently lack star-convexity. Similarly, robotics and autonomous vehicle problems do not satisfy this condition due to constraints like collision avoidance and traffic rules (Section C.2, revised paper). Relaxing this assumption extends the applicability of our methods to a broader range of real-world problems.
>
> **Challenges of Relaxation:** Without star-convexity, efficient exploration becomes more challenging due to the absence of global structure. Please note that in non-star-convex cases there is no line connecting the safe action to the optimal action within the action space. As a result, the agent may lack motivation to explore the optimal direction (unlike in the star-convexity scenario) to expand the estimated safe set toward the optimal direction. Hence, traditional exploration strategies may not perform well in unstructured decision sets and result in linear regret (please see Figure 1 in our paper).
>
> **Our Contribution:**  Removing the star-convexity assumption introduces substantial challenges, as key results from prior work, such as the fact that the **Optimism lemma** in Pacchiano et al. (2024), no longer holds in our setting. To address this issue, our method introduces a new bonus term, $g_t^\nu$ (Equation 4), which normalizes feature vectors to prioritize **uncertainty reduction** across all directions rather than focusing solely on expected rewards.  Using our bonus, we developed new theoretical tools, including a novel **Optimism lemma specifically designed for non-star-convex** decision sets (see **Lemmas 2, 5, 6**). Additionally, we proved that the **regret resulting from our new bonus remains sublinear** (see **Lemmas 4 and 7, and Theorem 1** in our paper). These contributions represent significant theoretical advancements that go beyond the introduction of the new bonus term and are critical for tackling non-star-convex settings effectively.
>
> Our approach not only addresses these challenges but also offers insights applicable to unconstrained bandit settings, potentially overcoming some weaknesses of UCB highlighted in [1]. We believe further research into this strategy is promising.
>
>
> Finally, we thank the reviewer for their helpful comments. If our response resolves your concerns, we kindly ask you to consider raising the rating of our work. We are happy to address any further questions during the discussion period
>
>
> [1] Russo, D., & Van Roy, B. (2014). Learning to optimize via information-directed sampling. *Advances in Neural Information Processing Systems*, 27.

---

> ### Author Response · Authors · 2024-11-25
>
> Dear Reviewer,
>
> As the author-reviewer discussion period is nearing its end, we would appreciate it if you could review our responses to your comments. This way, if you have further questions and comments, we can still reply before the author-reviewer discussion period ends. If our response resolves your concerns, we kindly ask you to consider raising the rating of our work. Thank you very much for your time and effort.

---

> > ### Comment · Reviewer_aNwp · 2024-11-26
> >
> > Thank you for your detailed responses.  I will maintain my current score.

---

### Official Review · Reviewer_H36G · 2024-10-31

**Soundness:** 2
**Presentation:** 2
**Contribution:** 3
**Rating:** 5
**Confidence:** 3

**Summary:**

This paper considers safety-constrained linear bandit problems under non-convex or discrete feature spaces. It introduces the Local Point Assumption, which assumes that the feature space is locally continuous or that the optimal action does not lie on the safety boundary. The paper develops the NCS-LUCB algorithm with a newly designed exploration bonus $b_t(a)$.

**Strengths:**

The algorithm employs double confidence intervals in conjunction with the exploration bonus $b_t(a)$, which seems interesting. Assumption 3 is also interesting as it captures the locally information. Additionally, the paper provides both upper and corresponding lower bounds, which strengthen the theoretical contributions.

**Weaknesses:**

In my view, the statements of the assumptions and their explanations are not clear and needs more elaboration. Please see the questions below for details. I am happy to discuss more in the rebuttal periods.

**Questions:**

- **Necessity and Correctness of Assumption 2**: To me the assumption 2 seems unnecessary, or could there be an error in its statement?  Here are some observations:
  - This action yields initial estimates of θ and γ as zeros.
  - In the toy example on non-convexity bias (line 403), the action set does not include $a_0 = [0,0]^{\top}$.
  - In the experiment described on line 491, the action set also does not include $a_0 = [0,0]^{\top}$.
  - In Lemma 7 (line 877), it is stated that $||\phi(a_t)||\geq \epsilon$ but  $a_0$ does not satisfy this condition.

- **Question about Assumption 3**
  - The first condition in Assumption 3 seems intuitively too strong. If we have the circle $||a_1||_2= 1$ in our action space, it results in a smaller safe circle within the action space. However, aren't two basis safe actions sufficient to estimate $\theta$ and $\gamma $ accurately? Are all the actions within this smaller safe circle necessary for the proofs and experiments? If so, is there a toy example that the algorithm won't work with only two basis safe actions?
  - Constraints on $L$ and $\tau$: When $L$ is very large (e.g., 100) and $\tau$ is very small (e.g., 0.01), how does the condition $L - \epsilon \leq 1$ hold in the second condition? Does this imply there are constraints on the relationship between $L$ and $\tau$? Could the authors provide more details or adjust the assumption accordingly?

- Could the authors elaborate on "What is the correct strategy?" Specifically, how is $b_t(a_1) = r^* - \tfrac{1}{3} = \tfrac{2}{3}$ derived? This seems different from equation (3).

- **Computational Issues:** I feel the proposed algorithm cannot be implemented to more general settings than discrete action space and have the following questions.
  - **Non-Discrete and Non-Convex Cases:** How does the proposed method computes the line 6 of Algorithm $a_t$ the non-convex non-discrete cases, such as when in action sets contains non-convex continuous regions?
  - **Convex Cases:** Even in convex regions, as $b_t(a)$ in (3) takes much complicated form, is the optimization problem in line 6 of Algorithm $a_t$ still a convex program as in linear bandit problem that can be approximate efficiently?

---

> ### Author Response · Authors · 2024-11-21
>
> We appreciate the reviewer’s thoughtful and constructive feedback.
>
> **Q.1**  To me the assumption 2 seems unnecessary, or could there be an error in its statement?
>
> **A.1**  Assumption 2 is adopted for notational simplicity but it is not needed. As noted in Remark 1 (see lines 216-219 in our revised paper), any non-zero safe starting action can be transformed into an equivalent problem satisfying Assumption 2 via a simple translation.
>
> Consider the following example: Assume we have access to a non-zero initial safe feature point $\phi_0$, which incurs a cost $\tau_0$. For an arbitrary point $\phi_1$, we have: $\phi_1 = \phi_0 + (\phi_1 - \phi_0)$. Since the cost function is linear with respect to feature points, we get:
>   $$
>   \text{cost}(\phi_1) = \text{cost}(\phi_0) + \text{cost}(\phi_1 - \phi_0) = \tau_0 + \text{cost}(\phi_1 - \phi_0)
>   $$
>
> Thus, the problem reduces to finding the cost of $\phi_1 - \phi_0$, with the safety threshold for the cost of $\phi_1 - \phi_0$ being $\tau - \tau_0$.
>
> **Q.2** This action yields initial estimates of $\theta$ and $\gamma$ as zeros.
>
> **A.2**  Although this action $a^0$ does not directly provide information about $\gamma$ and $\theta$, it does not imply that they are zero. Instead, the agent gains insight into $\gamma$ and $\theta$ by selecting actions near the safe point but oriented in different directions. Specifically, in a linear space, the Cauchy-Schwarz inequality establishes that any action $a$ satisfying $||a - a_{\text{safe}}||_2 \leq \frac{\tau - \tau_0}{\sqrt{d}}$ is guaranteed to be safe. By sampling these actions, the agent can systematically explore the environment while adhering to safety constraints.
>
>
> **Q.3** In the toy example on non-convexity bias (line 403), the action set does not include $a_0 = [0,0]^T$. In the experiment described on line 491, the action set also does not include $a_0 = [0,0]^T$.
>
> **A.3**  Thank you for pointing this out. We will revise the paper to include $a_0$ (see line 416 in our revised paper). However, as noted in A.1., the assumption (Assumption 2) is not needed, as the agent typically explores small features around it.
>
> **Q.4**   In Lemma 7, it is stated that $\|\phi(a_t)\| \geq \epsilon$, but $a0$ does not satisfy this condition.
>
> **A.4**  Please note that $a_t$ is the action selected by Algorithm 1. According to Lemma 7, Algorithm 1 ensures that $|\phi(a_t)| \geq \epsilon$, and therefore, it does not select $a_0$, as $a_0$ does not satisfy this condition.
>
>  **Q.5** The first condition in Assumption 3 seems intuitively too strong. If we have the circle $\|a_1\|_2 = 1$ in our action space, it results in a smaller safe circle within the action space. However, aren't two basis safe actions sufficient to estimate $\theta$ and $\gamma$ accurately? Are all the actions within this smaller safe circle necessary for the proofs and experiments? If so, is there a toy example that the algorithm won't work with only two basis safe actions?
>
>
> **A.5**  In order to obtain a small regret, it is important to be able to extend existing safe linear bandit methods. Hence, while we agree that $ d $ independent safe actions in $\mathbb{R}^d$ suffice for parameter estimation, extending existing safe linear bandit methods and achieving sub-linear regret bounds in such cases is highly challenging. For example, if the safe actions are strongly aligned along certain directions, learning and expanding the safe set in orthogonal directions can be significantly slower. This implies that it takes longer for the algorithm to include the optimal action in the estimated safe set. Therefore, a new algorithm design and analysis are required.
>
> Moreover, the $\iota$-neighborhood condition in Assumption 3 is essential. If the optimal action lies on the boundary of the safe set, the algorithm may fail to expand the safe set to include it, leading to linear regret. This requirement is reflected in our derived lower bound, where $\iota > 0$ is necessary.
>
> **Q.6** Constraints on $L$ and $\tau$: When $L$ is very large (e.g., 100) and $\tau$ is very small (e.g., 0.01), how does the condition $L - \epsilon \leq 1$ hold in the second condition? Does this imply there are constraints on the relationship between $L$ and $\tau$? Could the authors provide more details or adjust the assumption accordingly?
>
> **A.6**  There are no constraints on the relationship between $L$  and $\tau$. The reason for this is that any problem with $L \geq 1$ can be equivalently transformed to satisfy these conditions. Specifically, larger $L$ values can be handled through normalization by transforming $\phi$ to $\psi(a) = \frac{\phi(a)}{L}$ and adjusting $\tau$ to $\tau' = \frac{\tau}{L}$, ensuring $||\psi(a)||_2 \leq 1$.

---

> ### Author Response · Authors · 2024-11-21
>
> **Q.7** Could the authors elaborate on "What is the correct strategy?" Specifically, how is $ b_t(a_1) = r^* - \frac{1}{3} = \frac{2}{3}$  derived? This seems different from equation (3).
>
>
> **A.7**  In the toy example, the correct strategy to achieve sublinear regret is to revise the bonus $ b_t(.) $ for action $ a_1 $ to depend on the distance from $ a_1$ to the optimal point $ a_3$, rather than the distance from the safety boundary to the optimal point $ a_3$.
>
> Lines 445–454 (in our revised paper) explain that a proper bonus term should upper bound the distance from the optimal point $a_3$  to the nearest action in the estimated safe set, $a_1$ ( $||a_3 - a_1|| = \frac{1}{3}$). However, we did not mean that our bonus in Eq 3 equals 1/3. Lemma 2 demonstrates that our bonus upper bounds this distance, encouraging the algorithm to explore $a_1$, which, while not optimal compared to $a_2$, reduces uncertainty in the cost of the optimal action $a_3$.
>
>
> **Q.8** Computational Issues: I feel the proposed algorithm cannot be implemented to more general settings than discrete action space and have the following questions.
>
> - Non-Discrete and Non-Convex Cases: How does the proposed method compute the line 6 of Algorithm $a_t$ in non-convex, non-discrete cases, such as when the action sets contain non-convex continuous regions?
> - Convex Cases: Even in convex regions, as $b_t(a)$ in (3) takes a much more complicated form, is the optimization problem in line 6 of Algorithm $a_t$ still a convex program as in the linear bandit problem that can be approximated efficiently?
>
>
> **A.8** We agree that our algorithm may impose computational burdens, but this is quite common in non-convex settings, including star-convexity. When the feature space is convex, we should use more straightforward algorithms that focus on solving for the convex case. However, note that our setting when reduced to some special case, e.g, finite directions (as mentioned in Definition 12 of Pacchiano et al., 2024), our optimization step also has low complexity because $ g_t^\nu$ remains constant within a direction.
>
> Finally, we thank the reviewer for their helpful comments. If our response resolves your concerns, we kindly ask you to consider raising the rating of our work. We are happy to address any further questions during the discussion period

---

> ### Author Response · Authors · 2024-11-25
>
> Dear Reviewer,
>
> As the author-reviewer discussion period is nearing its end, we would appreciate it if you could review our responses to your comments. This way, if you have further questions and comments, we can still reply before the author-reviewer discussion period ends. If our response resolves your concerns, we kindly ask you to consider raising the rating of our work. Thank you very much for your time and effort.

---

> > ### Comment · Reviewer_H36G · 2024-11-26
> >
> > Thanks for the response, I'll keep the score for now and see if the other reviewer's concerns can be addressed. A minor thing is if Assumption 2 is not necessary, why not get rid of it?

---

> > > ### Author Response · Authors · 2024-11-27
> > >
> > > Thank you for your question. The purpose of Assumption 2 is to simplify the notation and maintain focus on the main ideas of our work. In the final version, we would omit Assumption 2 and state that, without loss of generality, we assume the safe starting action is $a^0 = 0$. As we mentioned in A.1, this assumption is valid because any non-zero safe starting action can be transformed into an equivalent problem satisfying this condition through a simple translation.

---

### Official Review · Reviewer_2wM7 · 2024-11-03

**Soundness:** 3
**Presentation:** 3
**Contribution:** 1
**Rating:** 3
**Confidence:** 4

**Summary:**

This paper studies the problem of stochastic linear bandits with instantaneous linear constraints, where the action set is not star-convex. Under an assumption on the action set that is weaker than star-convexity, they given an algorithm with $\tilde{O}(d\sqrt{T})$ regret.

**Strengths:**

The problem that this paper considers is a nice one (non-star-convexity in safe linear bandits). The requirement of star-convexity is a significant limitation in present works on safe linear bandits. The presentation is also good. Their use of visuals is quite effective.

**Weaknesses:**

Although I think the problem is important, this paper makes very little contribution to the literature. The core result is that they show $\tilde{O}(d\sqrt{T})$ regret under Assumption 3, which assumes that either (1) the constraint is not tight on the optimal point, or (2) a line segment in the direction of the optimal point and of sufficient length is within the action set. The results of Amani et al (2019) immediately show $\tilde{O}(d \sqrt{T})$ regret under condition (1). As for condition (2), the specific requirement is quite contrived and appears to be just a quirk of the analysis of Pacchiano et al (2021). I don't think this contribution alone is sufficient to justify another paper.

Additional points:
- The lower bound (Theorem 2) looks like it is essentially identical to that in Pacchiano et al (2021), and I would suggest adding discussion if/how it is different.
- I think that the related work section should probably be in the body of the paper (at least in part) and I would suggest adding [1].
- line 040: the paper writes that "the reward for an action in stochastic linear bandits is modeled as the inner product between a feature vector and unknown parameter." Really, it's that the *expected* reward is modeled as such.

[1] Gangrade, Aditya, Tianrui Chen, and Venkatesh Saligrama. "Safe Linear Bandits over Unknown Polytopes." The Thirty Seventh Annual Conference on Learning Theory. PMLR, 2024.

**Questions:**

- How is the lower bound (Theorem 2) different from Pacchiano et al. (2021)?
- I would suggest that the authors are more precise in how they discuss their contributions with respect to the claim of "first result for non-convex action sets." Previous works have considered non-convex sets, including those that are star-convex (Moradipari et al, 2021), as well as discrete sets (Pacchiano et al, 2021). I understand what the authors meant (i.e. first for non-star-convex sets and round-wise constraint satisfaction), but I suggest more precision to avoid confusion.

---

> ### Author Response · Authors · 2024-11-21
>
> We thank the reviewer for providing the constructive review.
>
> **Q.1**  Although I think the problem is important, this paper makes very little contribution to the literature. The core result is that they show $\tilde{O}(d\sqrt{T})$ regret under Assumption 3, which assumes that either (1) the constraint is not tight on the optimal point, or (2) a line segment in the direction of the optimal point and of sufficient length is within the action set. The results of Amani et al. (2019) immediately show $\tilde{O}(d\sqrt{T})$ regret under condition (1). As for condition (2), the specific requirement is quite contrived and appears to be just a quirk of the analysis of Pacchiano et al. (2021).
>
>
> **A.1** This is not true. The reason why this work is not a special case of Amani et al. (2019) under condition (1) is that their work assumes a **convex** decision space, which is quite restrictive and different from our Assumption 3. Hence, **Equation 5 in Amani et al. (2019) no longer holds**, rendering their **Lemma 1 and Lemma 2**—both essential for their sublinear regret—invalid in our context.
>
>
> Regarding condition (2), as shown in Figure 1, **directly applying the analysis from Pacchiano et al. (2021) to our setting results in linear regret in $T$**, as Lemma 4 (Optimism) in their work no longer holds in our problem setting. To address this issue, our method introduces a new bonus term, $g_t^\nu$ (Equation 4), which normalizes feature vectors to prioritize **uncertainty reduction** across all directions rather than focusing solely on expected rewards. This bonus is **key to proving Optimism for non-star convex decision sets** (see **Lemma 2** in our paper). Furthermore, we demonstrate that the **regret resulting from this new search method** remains **sublinear** (see **Lemma 4** in our paper).
>
> Please note that in non-star-convex cases there is no line connecting the safe action to the optimal action within the action space. As a result, the agent may lack motivation to explore the optimal direction (unlike in the star-convexity scenario) to expand the estimated safe set toward the optimal direction. However, in our method **even if the available action in the estimated safe set along the optimal direction seems suboptimal in terms of expected reward, $g_t^\nu$ still incentivizes the agent to take that action to reduce the uncertainty about the optimal direction.**
>
> **Q.2**  I don't think this contribution alone is sufficient to justify another paper.
>
> **A.2** We respectfully disagree with the view that our contributions are insufficient to justify this work. Removing the star-convexity assumption introduces substantial challenges, as key results from prior work, such as the fact that the **Optimism lemma in Pacchiano et al. (2024), no longer holds in our setting**. To address this, we developed new theoretical tools, including a **novel Optimism lemma** specifically designed for non-star-convex decision sets (see **Lemmas 2, 5, 6**). Additionally, we proved that the regret resulting from our new bonus **remains sublinear** (see **Lemmas 4 and 7, and Theorem 1** in our paper). These contributions represent significant theoretical advancements that go beyond the introduction of the new bonus term and are critical for tackling non-star-convex settings effectively.
>
>  Regarding practical importance, please note that in many practical scenarios, the star-convexity assumption does not hold. For instance, discrete action spaces, the VC example, and the smart building example discussed in our paper (see Lines 254-265, 697–711 in our revised paper) inherently do not satisfy the requirements for star-convexity. Similarly, robotics and autonomous vehicle problems do not satisfy star-convexity due to collision avoidance constraints and traffic rules (see Section C.2 in the revised paper). Therefore, our framework can be applied to these problems to achieve near-optimal performance. However, our framework is applicable to these cases, as the agent can learn by sampling from local neighborhoods around the initial safe actions.
>
>
> **Q.3** The lower bound (Theorem 2) looks like it is essentially identical to that in Pacchiano et al (2021), and I would suggest adding discussion if/how it is different?
>
> **A.3** Please note that Pacchiano et al’s lower bound does not apply to our case. The reason is that Pacchiano et al. 's lower bound is derived under a relaxed constraint, where the constraint is satisfied in expectation over the policy, rather than with high probability. As our approach requires the constraint to hold with high probability, their lower bound proof approach is not applicable to our setting because it would result in violation of our constraints.

---

> ### Author Response · Authors · 2024-11-21
>
> **Q.4**  I think that the related work section should probably be in the body of the paper (at least in part) and I would suggest adding [1].
>
> **A.4** Thank you for pointing this out. We have included a brief discussion of related works in the main text and added [1] to the final version (please see lines 136-150 in our revised paper).
>
>
> **Q.5**  I would suggest that the authors are more precise in how they discuss their contributions with respect to the claim of "first result for non-convex action sets."
>
> **A.5**  Thank you for pointing this out, and we will modify the phrasing in the final version to ensure clarity. Specifically, we have rephrased our contribution as follows: “In this work, we make the first attempt to design near-optimal safe algorithms for linear bandit problems with instantaneous hard constraints in non-star-convex and discrete spaces.”( see lines 111-113 in our revised paper).
>
>
> **Q.6** line 040: the paper writes that "the reward for an action in stochastic linear bandits is modeled as the inner product between a feature vector and unknown parameter." Really, it's that the expected reward is modeled as such.
>
> **A.6** Thank you for pointing this out. We have revised the phrasing to clarify that it is the expected reward that is modeled as the inner product between the feature vector and the unknown parameter (see lines 40-42 in our revised paper).
>
> Finally, we thank the reviewer again for the helpful comments and suggestions for our work. If our response resolves your concerns to a satisfactory level, we kindly ask the reviewer to consider raising the rating of our work. Certainly, we are more than happy to address any further questions that you may have during the discussion period.

---

> ### Author Response · Authors · 2024-11-25
>
> Dear Reviewer,
>
> As the author-reviewer discussion period is nearing its end, we would appreciate it if you could review our responses to your comments. This way, if you have further questions and comments, we can still reply before the author-reviewer discussion period ends. If our response resolves your concerns, we kindly ask you to consider raising the rating of our work. Thank you very much for your time and effort.

---

> ### Comment · Reviewer_2wM7 · 2024-11-25
>
> Thank you for your detailed response. I am sticking with my score for now, but I wanted to provide more detail on my initial comments.
>
> In my discussion of Amani (2019), I was just pointing out that their exploration-exploitation algorithm will suffice under Assumption 3.1. Indeed, Amani (2019) showed that when the constraint is loose on the optimal action, it is possible to use an exploration-exploitation algorithm to get $\tilde{O}(d\sqrt{T})$ regret. Although that paper uses the assumption of convex action set, it is sufficient to assume that there is a set of actions that is initially known to be safe and is full-dimensional. It seems that you are pointing out the case where this set is not full-dimensional. However, your Assumption 3 ensures that for every action, there is an action in that direction that is known to be safe, so the dimension of this initial safe set will be the same as the action set. Therefore, if the initial safe set is not full-dimensional, then you can always work in the lower-dimensional space where it is full dimensional (without loss of generality).
>
> In my discussion of Pacchiano (2021), I was not saying that their theorems are directly applicable, it's just that the present paper only seems to show that the approach used by Pacchiano can handle sets that are "almost" star-convex. Indeed, instead of requiring a line segment between the origin and optimal action to be in the set, Assumption 3.2 requires a "partial" line segment to be in the set. Precisely, Assumption 3.2 requires that this partial line segment extends from the optimal point some nonzero distance. Under this assumption, wouldn't it be sufficient to just explore until this line segment intersects with the safe set and then play the algorithm from Pacchiano for the remaining rounds?

---

> ### Author Response · Authors · 2024-11-26
>
> Thank you for your clarification. **Combining** the methods of **Amani** et al. and **Pacchiano** et al. will **not** achieve the desired performance in our non-convex setting because applying Amani et al.'s exploration strategy **fails** to guarantee **sufficient expansion** toward the optimal point.
>
> In the pure exploration phase of Amani et al., their method relies on random sampling from a sphere within the initial safe set (see Remark 1 and Algorithm 1 in their work). This uniform sphere ensures that random sampling leads to an even expansion of the estimated safe set in all directions.
>
> However, in our non-convex setting, the geometry of the initial safe set can be highly irregular. This irregularity causes the density of sampled points to be biased toward suboptimal directions. Consequently, applying Amani’s method could lead to significant expansion in suboptimal directions while minimally expanding toward the optimal direction.
> Specifically, the critical Lemmas 1 and 2 in Amani’s work no longer hold in our setting. Hence, we **cannot** guarantee that the optimal point (or the optimal line segment in case 2) will be **included in the estimated safe set within the $T_\Delta$ time bound specified in their Lemma 2.**
>
> For example, consider a simple scenario in $\mathbb{R}^2$. Suppose the initial safe set contains three points: $a_1$ and $a_2$ on the $x$-axis, and $a_3$ on the $y$-axis. Random sampling from this set would result in twice the expansion along the $x$-axis compared to the $y$-axis. This imbalance becomes even more severe in complex, high-dimensional scenarios.
>
> In contrast, our method incorporates the term $g_t^\nu$ in the exploration bonus, explicitly focusing on reducing uncertainty in all directions. This deliberate design ensures a more balanced and effective expansion of the safe set, allowing us to achieve sublinear regret.
>
> We appreciate your insightful comments and welcome any further questions or concerns, as they help us highlight the critical contributions of our work.

---

> > ### Comment · Reviewer_2wM7 · 2024-12-02
> >
> > Thank you for the detailed response.
> >
> > I apologize, but I do not follow your reasoning as to why we can't get $O(d \sqrt{T})$ in your setting by first using iid exploration to sufficiently grow the conservative set, and then using the Pacchiano algorithm for the remaining rounds. It seems that your main argument is that we can't use iid exploration to grow the constraint set. I do not think this claim is correct. All that we need for the iid exploration is to sample $x_t \sim \mathcal{U}$ such that $\lambda_{min} (\mathbb{E}[x_t x_t^T]) > 0$ because Lemma 1 in Amani then ensures that $|| x_* ||_{\Lambda^{-1}} \leq C /\sqrt{T'}$ w.h.p., where $T'$ is the duration of the exploration phase. In your setting, this can be guaranteed by choosing $\mathcal{U}$ to be uniform over $\mathcal{A}^{\tau/\sqrt{d}}$. Then, thanks to Assumption 3, either $\lambda ( \mathbb{E} [x_t x_t^T]) > 0$, or the action set is in a lower dimensional subspace and we can transform the problem to ensure $\lambda ( \mathbb{E} [x_t x_t^T]) > 0$. *In your example, uniformly sampling $a_1, a_2, a_3$ will ensure that $\lambda ( \mathbb{E} [x_t x_t^T]) > 0$ and therefore the exploration will be effective.* Then, it follows that $\frac{\tau}{\tau + \iota} x^* \in \mathcal{A}_t^{RLS}$ when $T' \geq \iota^2 \beta^2$, which will only result in an additional $d \log(T)$ term in the regret due to the exploration. This ensures that there is a line segment from the conservative set to the optimal action and therefore I believe that the Pacchiano algorithm can be applied to ensure $O(d \sqrt{T})$ regret for the remaining rounds.
> >
> > In any case, my primary concern was the technical novelty with respect to prior work. After reading the comments by the authors, it still seems to me that the paper's contribution does not go very far beyond showing that the Pacchiano algorithm works in a slightly relaxed setting, i.e. when the action set is "partially" star-convex. I appreciate the authors efforts in detailing their approach of  "Normalization for Reducing Uncertainty" in the global comment, but I don't see how this is a novel conceptual viewpoint. For example, the idea of "encouraging the exploration of actions that may not appear optimal but reduce uncertainty about the optimal action" sounds conceptually very similar to how UCB balances exploration and exploitation.

---

> > > ### Author Response · Authors · 2024-12-03
> > >
> > > We thank the reviewer for their feedback.
> > >
> > > We understand your reasoning regarding the potential use of i.i.d. exploration to grow the conservative set and then apply the Pacchiano algorithm. However, the main challenge lies in constructing a uniform $\mathcal{U} \subset \mathcal{A}^{\tau/\sqrt{d}}$, which may not be tractable in our non-convex problem setting.
> > >
> > > By "uniform," we mean that $\mathcal{U}$ contains an equal number of points in each direction, ensuring that the likelihood of sampling in the optimal direction is the same as in any other direction. This uniformity guarantees that i.i.d. sampling from $\mathcal{U} \subset \mathcal{A}^{\tau/\sqrt{d}}$ results in an even expansion of the estimated safe set.
> > >
> > > In Amani et al. (2019), constructing such a $\mathcal{U}$ is straightforward because the action space is convex, allowing for a uniform distribution over a small sphere within the initial safe set. However, under Assumption 3 in our setting, the geometry of the initial safe set is highly irregular and **non-convex**. This irregularity makes it **intractable** to construct a uniform $\mathcal{U} \subset \mathcal{A}^{\tau/\sqrt{d}}$ over the estimated safe set, particularly in continuous spaces.
> > >
> > > Moreover, if the initial safe set $\mathcal{U} \subset \mathcal{A}^{\tau/\sqrt{d}}$ is **not uniform**, i.i.d. sampling becomes problematic. For example, if the density of points in the initial safe set along the optimal direction is significantly lower than in other directions, even uniform sampling from $\mathcal{U}$ may result in a **low likelihood** of obtaining samples that expand the conservative set in the optimal direction. Consequently, the safe set may fail to expand sufficiently in the optimal direction, undermining the effectiveness of the exploration phase and preventing the Pacchiano algorithm from achieving $O(d \sqrt{T})$ regret.
> > >
> > >
> > > Also, in our example, the action set is $\mathcal{U} = \{a_1, a_2, a_3\}$, where $a_1$ and $a_2$ lie on the $x$-axis, and $a_3$ lies on the $y$-axis. If we uniformly sample from $\mathcal{U}$, then on average, for every three samples, one is $a_1$, one is $a_2$, and one is $a_3$. This results in $a_1$ and $a_2$ being selected twice as often as $a_3$. Consequently, two-thirds of our samples are concentrated along the $x$-axis, and only one-third are on the $y$-axis. This sampling imbalance leads to the safe set expanding much more rapidly along the $x$-axis than along the $y$-axis.
> > >
> > >  Our proposed solution addresses this issue by focusing on directional search. While traditional UCB is designed to prioritize points that appear promising in terms of reward [2], our term $g_t^\nu$ explicitly promotes exploration across different directions.  Even if points in the optimal direction within the initial safe set seem suboptimal (e.g., their norm is significantly smaller compared to points in suboptimal directions included in the initial safe set), $g_t^\nu$ compels the agent to explore that direction. This approach ensures sufficient exploration, even when the initial safe set does not appear promising along the optimal direction. Please see the new simulation results provided in **Appendix P and Figure 7**.
> > >
> > > In summary, while the approach you suggest works well in star-convex settings, it does not extend to our non-star-convex action set. The irregular geometry of our problem requires a different strategy to ensure adequate exploration in the optimal direction, which is what our proposed method addresses.
> > >
> > > [2] Russo, D., & Van Roy, B. (2014). Learning to optimize via information-directed sampling. Advances in Neural Information Processing Systems, 27.

---

> ### Author Response · Authors · 2024-12-02
>
> Dear Reviewer,
> As the author-reviewer discussion period is nearing its end, we would appreciate it if you could review our responses to your comments. This way, if you have further questions and comments, we can still reply before the author-reviewer discussion period ends. Thank you for your time and effort.

---

### Author Response · Authors · 2024-11-21
**Global Comment**

We appreciate the thoughtful feedback from all the reviewers and the opportunity to clarify the contributions of our work. We have revised our paper to address the comments from the reviewers. Below we highlight a few of the key issues that reviewers have raised.

**Difference and Issue in Star-Convexity in Existing Methods:**
Pacchiano et al. (2024) utilize the assumption of star-convexity to facilitate exploration toward the optimal action. Star-convexity ensures that any line segment from a central point to a point in the set remains within the set. This assumption is useful because it guarantees that within the estimated safe set, there is always an action in the direction of the optimal action that appears promising to the agent. Motivated by this, the agent is inclined to explore along the optimal direction. By adjusting the UCB bonus, Pacchiano et al. (2024) encourage the agent to select actions along this direction, confidently expanding the estimated safe set toward the optimal action.

However, many practical scenarios violate the star-convexity assumption. For instance, discrete action spaces, the VC example, and the smart building example discussed in our paper (see Line 254-265, 697-711 in the revised paper) do not satisfy the star-convexity assumption. Similarly, real-world challenges such as autonomous vehicle control also fail to meet this assumption. In autonomous vehicle control, collision avoidance constraints result in non-convex and non-star-convex decision sets (refer to Section C.2 in the revised paper).

In such cases, there is no line connecting the safe action to the optimal action within the action space. As a result, the agent may lack motivation to explore the optimal direction (unlike in the star-convexity scenario), which could lead to linear regret if not properly addressed, as illustrated in Figure 1 of our paper. Our solution addresses this issue by enabling effective exploration and safe set expansion in non-star-convex decision spaces.

**Our Novel Contribution ( Normalization for Reducing Uncertainty):**
Our work introduces a novel exploration strategy that addresses these limitations without relying on star-convexity. We design an exploration bonus, $g_t^\nu$, which normalizes feature vectors to prioritize **uncertainty reduction** across all directions rather than focusing solely on expected rewards. This encourages the exploration of actions **that may not appear optimal but reduce uncertainty about the optimal action**. Without $g_t^\nu$, the agent might focus excessively on actions with high immediate rewards, potentially leading to suboptimal exploration and linear regret. **Even if the available action in the estimated safe set along the optimal direction seems suboptimal in terms of expected reward, $g_t^\nu$ still incentivizes the agent to take that action to gather information about the optimal direction.**

This “information-driven” approach enables us to prove **Optimism** for our problem (see **Lemma 2**), whereas existing methods, such as Pacchiano et al. (2024), fail to satisfy Optimism in our setting. A potential **concern** could be whether this information-driven search results in **linear regret**.  As demonstrated in our updated Figure 1, **Pacchiano et al.'s method** indeed incurs **linear regret** in our problem setting. However, our carefully designed $g_t^\nu$ ensures a **regret bound** of $\widetilde{\mathcal{O}}\left( d \frac{ \tau}{\epsilon \iota} \sqrt{T} \right)$, adding only a $\frac{1}{\epsilon \iota}$ factor compared to convex cases(Please see **Lemma 4** in our paper). The lower bound in **Theorem 2** confirms that this dependency is **unavoidable**.


**Advancing the Field of Safe Bandits:**
Our approach contributes to the field in several significant ways:

**1. Enhanced Exploration:** The normalization in $ g_t^\nu $ promotes exploration in all directions, enabling the agent to acquire valuable information that traditional UCB methods might miss,  specifically in safe bandits settings where the constraints make the action set uneven or incoherent/unstructured. This improves the practicality of safe bandit algorithms in real-world scenarios with non-star-convex decision sets (e.g., discrete cases, the VC example in Lines 254-265, and the smart building problem in Lines 697–711).

**2. New Theoretical Results for safe constraints at each step:**   We demonstrated that our search method achieves sublinear regret while remaining safe with high probability at each step of the algorithm, as stated in Theorem 1. This advances the field of safe bandits by addressing challenges in unstructured decision spaces, where existing methods either fail to provide guarantees or only apply to significantly simpler settings. Notably, applying their results to our case would lead to linear regret (please see Figure 1), as they are not designed to handle the complexity of our problem.

---

> ### Author Response · Authors · 2024-11-21
>
> 3. **New Insights for UCB Methods:** Our findings highlight a new avenue for enhancing UCB-based methods by incorporating normalization to focus on information gain (or uncertainty reduction). This approach is not only useful for bandits with safe constraints, but could inspire further research into more effective exploration strategies in both safe and standard bandit settings.
>
> As described above, our contributions are substantial and open new research avenues. We hope this clarification addresses the reviewers' concerns. We again thank reviewers for their consideration, and welcome any further questions or discussions.

---

### Meta-Review · Area_Chair_K72z · 2024-12-21

**Metareview:**

This paper studies safe linear bandits in non-convex feature spaces. Compared to existing algorithms for safe linear bandits, the new method does not rely on the star-convexity assumption required by prior works.

The main weakness raised by the reviewers is the lack of novelty. It is unclear how much new insights this paper brings over prior work. This paper could be significantly improved by having a more detailed comparison with existing work on safe linear bandits in terms of the technical aspects. Moreover, the new algorithm relies on instance-dependent parameters. More discussion on how to handle cases when those parameters are unknown could be beneficial.

Given the high standards of ICLR and the weakness mentioned above, I would recommend rejecting this paper.

**Additional Comments On Reviewer Discussion:**

The reviewers raised concerns regarding dependence of the new algorithm on instance-dependent parameters, clarity and necessity of the assumptions, as well as the lack of technical novelty compared to existing work on safe linear bandits. Although the authors provided responses which addressed some of those concerns, concerns regarding the lack of technical novelty remain.

---

### Decision · Program_Chairs · 2025-01-22

Reject